# Improving Code Translation Correctness and Efficiency with Multi-Perspective Exploration and Difference-Aware Selection

## Abstract

While large language models (LLMs) have greatly advanced the functional correctness of automated code translation systems, the runtime efficiency of translated programs has received comparatively little attention. With the waning of Moore's law, runtime efficiency has become as critical as functional correctness in evaluating program quality. Our preliminary study reveals that LLM-translated programs often run slower than human-written ones, and this issue cannot be remedied through prompt engineering alone. Therefore, our work proposes SWIFTTRANS, a code translation framework comprising two key stages: (1) **Multi-Perspective Exploration**, where *MpTranslator* leverages parallel in-context learning (ICL) to generate diverse translation candidates; and (2) **Difference-Aware Selection**, where *DiffSelector* identifies the optimal candidate by explicitly comparing differences between translations. We further introduce *Hierarchical Guidance* for MpTranslator and *Ordinal Guidance* for DiffSelector, enabling LLMs to better adapt to these two core components. To evaluate the runtime efficiency of programs, we extend existing benchmarks, CodeNet and F2SBench, with efficiency-critical test cases and maximum runtime constraints on translated programs. We also introduce SWIFTBENCH, a new benchmark designed to evaluate whether translation models can improve the efficiency of programs when the source code exhibits inefficiencies. Experimental results across all three benchmarks show that SWIFTTRANS achieves consistent improvements in both correctness and efficiency. Notably, SWIFTTRANS built on Qwen2.5-7B surpasses current state-of-the-art models such as GPT-5 and training-based F2STrans (Zhang et al., 2025b).

## 1 Introduction

Code translation, the task of converting code from a source programming language (*e.g.,* C) to a target language (*e.g.,* Python), is vital in software engineering scenarios like legacy system migration and cross-platform development (Mossienko, 2003). The rise of large language models (LLMs) has introduced a new paradigm for code translation. Unlike earlier methods relying on handcrafted features (Zhong et al., 2010) or intricate deep architectures (Chen et al., 2018), LLMs can perform preliminary translation through simple prompt learning (Yan et al., 2023). This has attracted increasing research attention on enhancing the functional correctness of code translated by LLMs, and significant progress has been made (Zhang et al., 2025a). For example, Yang et al. (2024); Ibrahimzada et al. (2025b) leverage compilers to detect translation bugs, enabling targeted repairs by LLMs.

However, according to the ISO/IEC 25010 guidelines (ISO/IEC25010, 2011), program quality includes not only functional correctness but also non-functional attributes such as efficiency. Despite progress in **Functional Correctness** (Yin et al., 2024; Ibrahimzada et al., 2025a), **Runtime Efficiency**—a crucial aspect of program performance—has received little attention in prior work. To address this gap, we conduct a preliminary investigation and present two key findings: (1) *LLM-translated code typically exhibits lower efficiency than human-written code in the target language*, as shown in Fig. 1 (a). One major reason is that LLMs tend to replicate the logic and structure of the source code (Zhang et al., 2025b). Although such replication reduces the risk of errors, it also perpetuates any inefficient coding constructs present in the source code and neglects target language-specific optimizations, such as C pointers or Python's built-in functions. (2) *Ensuring both correctness and*

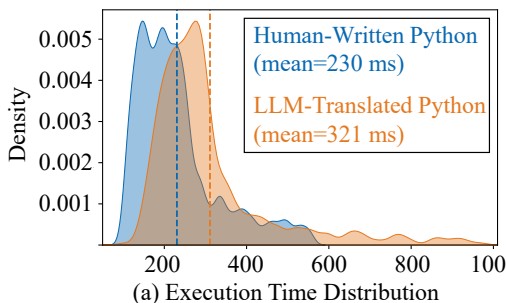 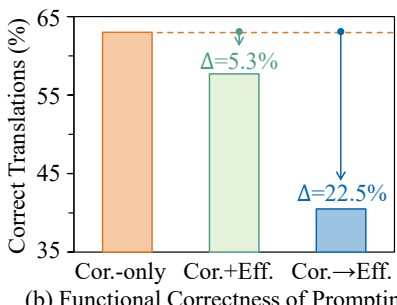

(a) Execution Time Distribution

(b) Functional Correctness of Prompting

Figure 1: Challenges in runtime efficiency of LLM-translated code, shown on C-to-Python translation from F2SBench (Zhang et al., 2025b) with Qwen3-Next-80B (Qwen, 2025). (a) LLM-translated programs generally run slower than human-written ones. (b) This issue is hard to address, as prompt engineering strategies—such as prompts that additionally emphasize efficiency ("Corr.+Eff.")  or employ post-hoc optimization ("Corr.→Eff.")—can improve efficiency but often reduce functional correctness relative to correctness-only prompts ("Corr.-only").

*efficiency in translated code remains challenging*, as shown in Fig. 1 (b). Straightforward solutions, like complex prompts or post-hoc optimization modules, often improve efficiency at the cost of correctness due to increased complexity.

Our work introduces SWIFTTRANS, a code translation framework designed to ensure both correctness and efficiency. SWIFTTRANS first employs a **Multi-Perspective Translator** (MpTranslator) to generate diverse translation candidates from the source code, and then applies a **Difference-Aware Selector** (DiffSelector) to identify the optimal one. MpTranslator draws on diverse, multi-scale demonstrations, which improves translation quality and diversity compared to traditional repeated sampling (Brown et al., 2024). Through hierarchical guidance training, MpTranslator learns to produce outputs that range from conservative (correctness-first) to optimized (efficiency-aware) translations, enabling adaptation to tasks of varying complexity. Serving as a pairwise LLM-as-a-judge, DiffSelector performs fine-grained comparisons between translation candidates, considering both correctness and efficiency. It employs an efficient linear-time selection strategy, inspired by bubble sort, to evaluate all candidates. Finally, we introduce ordinal-guidance training to enhance DiffSelector's accuracy and robustness to candidate order.

To support a comprehensive evaluation of code translation models, we introduce enhanced benchmarks that assess not only functional correctness but also runtime efficiency. Current benchmarks, such as CodeNet (Puri et al., 2021) and F2SBench (Zhang et al., 2025b), typically contain only limited, simple test cases that emphasize functional correctness. To address this limitation, we augment these benchmarks with manually curated, efficiency-critical test cases and corresponding maximum runtime constraints on translated programs. Moreover, we propose SWIFTBENCH, a new benchmark that incorporates source programs with intentionally embedded inefficiencies, such as redundant computations or suboptimal algorithmic choices. This design evaluates whether translation models can eliminate inefficiencies in translated code without compromising functional correctness. Additionally, SWIFTBENCH is regularly updated to mitigate the risk of data contamination.

Extensive experiments on CodeNet, F2SBench, and SWIFTBENCH show that SWIFTTRANS consistently surpasses existing methods in both functional correctness and runtime efficiency. For example, across translation tasks among C, C++, Go, Java, and Python, SWIFTTRANS with Qwen2.5-7B outperforms GPT-5 (OpenAI, 2025) and training-based approaches, such as F2STrans (Zhang et al., 2025b). Ablation studies further validate the effectiveness of both MpTranslator and DiffSelector.

Our key contributions are summarized as follows:

- To our knowledge, we are the first to systematically highlight and address efficiency deficits in LLM-based code translation, for which we propose the SWIFTTRANS framework.

- We extend existing benchmarks and develop a new benchmark, SWIFTBENCH, to support the evaluation of both correctness and efficiency.

- Experiments across diverse benchmarks and programming languages show that our approach significantly improves the quality of translated code compared to various baselines.

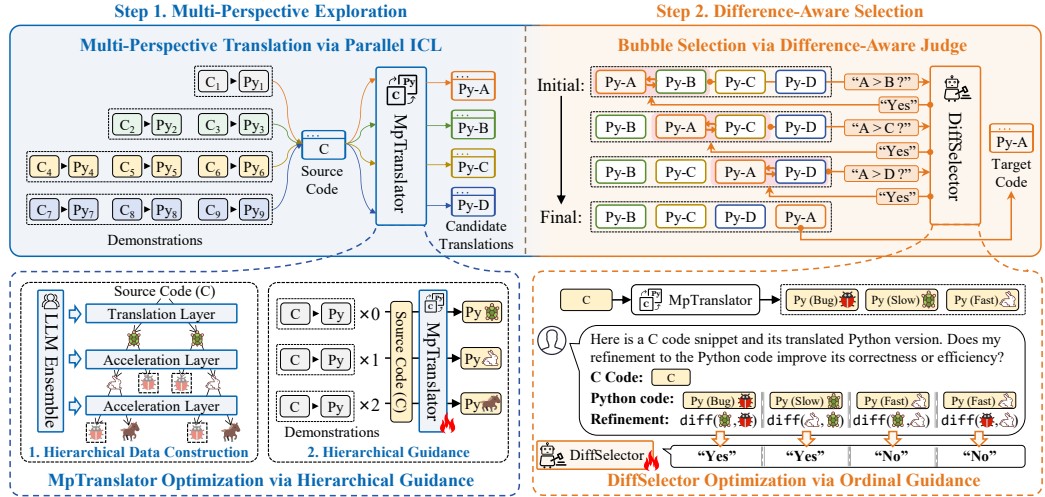

Figure 2: Overview of our SWIFTTRANS. Taking C-to-Python translation as an example, MpTranslator first generates diverse candidates through parallel ICL, and DiffSelector applies a difference-aware judging strategy with bubble selection to identify the most accurate and efficient translation. We introduce hierarchical guidance for MpTranslator and ordinal guidance for DiffSelector to better adapt LLMs to these two core components.

## 2 METHODOLOGY

As shown in Fig. 2, given a source code snippet, our SWIFTTRANS framework first applies the *Multi-Perspective Exploration* to generate a diverse set of candidate translations, and then selects the optimal one through *Difference-Aware Selection*. In this process, LLMs provide critical support for SWIFTTRANS's two core components: *MpTranslator* and *DiffSelector*. We optimize LLMs specifically for these two components, enabling lightweight open-source LLMs (*e.g.,* Qwen2.5-3B) to match or even surpass the performance of powerful LLMs like GPT-5.

### 2.1 MULTI-PERSPECTIVE EXPLORATION

This subsection first describes the multi-perspective translation mechanism of MpTranslator, which leverages parallel in-context learning (ICL) to generate diverse candidates. Next, it details the hierarchical guidance strategy used to optimize MpTranslator.

#### 2.1.1 MULTI-PERSPECTIVE TRANSLATION VIA PARALLEL ICL

Traditional repeated sampling approaches (Brown et al., 2024) generate multiple outputs by issuing identical prompts to the LLM. However, constrained by fixed inputs, these outputs remain confined to a narrow semantic space (Wang et al., 2024b).

To overcome this limitation, MpTranslator leverages parallel ICL to encourage diversity in candidate translations. Specifically, for a source code snippet $src$, MpTranslator first randomly constructs $m$ sets of demonstrations from a large demonstration library $\mathcal{C}$. Each set contains a random number (ranging from 0 to $K$) of demonstrations. The demonstration library $\mathcal{C}$ is derived from hierarchical guidance data, which will be discussed in the following section. MpTranslator then generates candidate translations in parallel, conditioned on each demonstration set. Compared to vanilla repeated sampling, MpTranslator offers two key advantages. First, ICL generally elicits significantly higher-quality responses from LLMs than direct prompt learning. Second, parallel ICL can explicitly induce LLMs to explore diverse responses by varying the provided context.

#### 2.1.2 MPTRANSLATOR OPTIMIZATION VIA HIERARCHICAL GUIDANCE

To enhance the adaptability of lightweight, open-source LLMs to the MpTranslator, we employ the hierarchical guidance strategy grounded in instruction fine-tuning (IFT). Standard IFT optimizes LLMs

via next-token prediction, improving their capacity to follow task-specific instructions. However, its direct application to MpTranslator faces two limitations: First, traditional IFT uses only the source code as input, while MpTranslator requires additional demonstrations as context during inference. This input inconsistency between training and inference can degrade model performance. Second, IFT learns from a single ground-truth response, which can lead to diversity collapse (Dang et al., 2025) in the model's outputs. To address these issues, we propose a hierarchical guidance training.

**Hierarchical Data Construction.** We construct multi-level target code from source code collected on online platforms (*e.g.,* Codeforces). Lower levels correspond to functionally correct but slower implementations, while higher levels represent progressively optimized, faster versions.

Specifically, an ensemble of powerful LLMs (*e.g.,* Qwen2.5-Coder-32B, gpt-oss-20B) first generates initial translations focusing on functional correctness, with each LLM contributing one candidate. The ensemble then iteratively edits and accelerates these translations for up to $n$ rounds. A code compiler, leveraging online platform-provided test cases, filters out translations that are functionally incorrect or fail to achieve runtime improvement. Through this process, the ensemble contributes diverse strategies for translation and acceleration.

From each level, we randomly sample one code snippet, ensuring that each level exhibits at least a 10% speedup over the previous one. The source code $src$ and its most optimized translation $tgt^n$ are stored in the demonstration library $\mathcal{C}$. In addition, $src$ and its hierarchical translations $\{tgt^0, ..., tgt^n\}$ form our hierarchical training dataset, where $tgt^0$ is the initial functionally correct translation, and $tgt^{1...n}$ are increasingly optimized variants.

**Hierarchical Guidance.** We use the constructed hierarchical data to train LLMs, yielding the final MpTranslator. First, for each source code $src$ and its target code $tgt^t$ at optimization level $t$, we randomly sample a subset $\mathcal{D}^t$ from the demonstration library $\mathcal{C}$, with the size of $\mathcal{D}^t$ set to $t$ to match the optimization level. For the base level $tgt^0$, which focuses solely on correctness, we set $D^0 = \emptyset$. We then train the model with demonstrations as context, with the loss defined as follows:

$$\mathcal{L}_{\text{hg}}(src, \mathcal{D}^0, tgt^0, \ldots, \mathcal{D}^n, tgt^n) = -\frac{1}{n+1} \sum_t \sum_i \log p\left(tgt_i^t \mid \mathcal{D}^t, src, tgt_{<i}^t\right), \quad (1)$$

where $tgt_i^t$ denotes the $i$-th token of $tgt^t$, and $tgt_{<i}^t$ represents the preceding token sequence.

This hierarchical guidance provides three key advantages: (1) Training with demonstrations as context ensures consistency between training and inference. (2) Learning from multiple translations per source mitigates the diversity collapse (Dang et al., 2025) inherent in standard IFT. (3) Linking the size $t$ of demonstration set $\mathcal{D}^t$ to the optimization level teaches the model to produce conservative translations under sparse context and increasingly efficient translations with richer context, thereby adapting flexibly to tasks of varying difficulty.

## 2.2 DIFFERENCE-AWARE SELECTION

This subsection first introduces the workflow of the DiffSelector component, which employs a difference-aware judge to evaluate translation quality and utilizes a bubble-selection strategy for optimal candidate selection. Next, it presents ordinal guidance, which optimizes DiffSelector to achieve greater accuracy and robustness.

### 2.2.1 BUBBLE SELECTION VIA DIFFERENCE-AWARE JUDGE

The LLM-as-a-judge strategy is commonly used to select the optimal candidate from multiple outputs generated by LLMs (Zheng et al., 2023). However, since translations originate from the same source code, their differences are often subtle, sometimes limited to only a few tokens. Distinguishing such minor variations is challenging for LLMs.

To address this, we introduce DiffSelector, a difference-aware selector designed to facilitate fine-grained discrimination among similar translations. DiffSelector adopts a pairwise comparison strategy, evaluating two translations at a time. Unlike conventional methods, it treats one translation as a modified version of the other, explicitly highlighting their differences to support more accurate judgments. As illustrated in Fig. 2, the diff($tgt_1$, $tgt_2$) operation shows the modifications from $tgt_1$ to $tgt_2$ in unified diff format, computed using GNU diff.

A straightforward use of DiffSelector is to compare every pair of candidate translations and select the best one, which requires $\mathcal{O}(n^2)$ comparisons for $n$ candidates. To improve efficiency, we draw inspiration from the bubble sort algorithm, in which elements are compared and swapped based on pairwise evaluations. Specifically, we utilize DiffSelector as the pairwise comparator and treat candidate translations as elements to be sorted by quality. As shown in Fig. 2, we first compare "Py-A" and "Py-B", retain the better one, and then compare it against the third candidate "Py-C". The process repeats sequentially until all candidates have been evaluated. In this way, DiffSelector identifies the best translation in a single pass with only $n - 1$ comparisons, achieving $\mathcal{O}(n)$ complexity.

### 2.2.2 DIFFSELECTOR OPTIMIZATION VIA ORDINAL GUIDANCE

We further enhance DiffSelector through ordinal guidance, which leverages the inherent ranking of translation quality: efficient and correct translations $\succ$ slower correct translations $\succ$ incorrect translations. Firstly, MpTranslator generates multiple candidate translations from source code $src$ collected on online platforms. Based on compiler feedback, we then select two target translations of different quality, denoted as $tgt^+$ and $tgt^-$. For example, $tgt^+$ is correct and efficient code, while $tgt^-$ is correct but less efficient. Given the source code $src$ and the two targets, we propose a bi-judge loss that trains the LLM to judge their relative quality bidirectionally, *i.e.,* whether $tgt^+$ constitutes an improvement over $tgt^-$ and vice versa. The loss function is defined as:

$$\mathcal{L}_{\text{og}}(src, tgt^+, tgt^-) = -\frac{1}{2} \left[ \log p \left( \text{'Yes'} \mid src, tgt^+ \succ tgt^- \right) + \log p \left( \text{'No'} \mid src, tgt^- \succ tgt^+ \right) \right] \quad (2)$$

where "Yes" and "No" denote the ground-truth responses for the relative quality between $tgt^+$ and $tgt^-$. This bi-judge design mitigates sensitivity to candidate order (Zheng et al., 2023) in the prompt.

## 3 EXPERIMENTS

### 3.1 BENCHMARK CONSTRUCTION

**Extension of Existing Benchmarks.** Current benchmarks, such as CodeNet (Puri et al., 2021) and F2SBench (Zhang et al., 2025b), primarily focus on functional correctness but offer little support for efficiency evaluation, due to two main limitations: (1) The test cases are too simple to reveal runtime performance differences. For example, $\mathcal{O}(n^2)$ and $\mathcal{O}(n)$ implementations often show negligible runtime differences when $n = 1$. (2) The lack of baseline execution times prevents reliable efficiency evaluation. To address these limitations, we manually augment each sample in CodeNet and F2SBench with (i) ten efficiency-critical test cases and (ii) the maximum baseline execution time derived from conservative translations. Annotation is performed by three independent teams, each consisting of 20 experienced software professionals. From the collected annotations, we select the ten most diverse and challenging test cases for each sample. For runtime evaluation, we annotate multiple conservative translations and adopt the slowest execution time among them as the reference.

**Construction of SWIFTBENCH.** Beyond extending existing benchmarks, we introduce a new benchmark, SWIFTBENCH. Similar to CodeNet and F2SBench, SWIFTBENCH collects source code from online platforms, such as Codeforces, and provides both efficiency-critical test cases and a baseline execution time of target code. Distinctively, each source program in SWIFTBENCH contains intentional efficiency issues, such as redundant computations or suboptimal algorithms. This design reflects real-world scenarios, where source code quality is often unpredictable. Consequently, SWIFTBENCH evaluates whether translation models can improve inefficient input programs. To further reduce benchmark leakage in LLM evaluation (Xu et al., 2024), SWIFTBENCH is updated quarterly with programming problems recently released on online platforms. The current version covers problems released from June to August 2025. App. B provides additional details about the SWIFTBENCH benchmark.

### 3.2 EXPERIMENTAL SETTINGS

#### 3.2.1 IMPLEMENTATION DETAILS

In the multi-perspective translation via parallel ICL, we set the number of demonstration sets $m$ to 10, with each set containing up to $K = 3$ examples. For the hierarchical data construction, the

LLM ensemble consists of DeepSeek-Coder-V2-Lite-Instruct-16B, gpt-oss-20B, and Qwen3-Coder-30B-A3B-Instruct, with the code acceleration depth $n$ fixed at 3. Our experiments cover translation among five programming languages: C, C++, Go, Java, and Python, yielding a total of 20 translation scenarios. Both the hierarchical guidance for MpTranslator and ordinal guidance for DiffSelector utilize approximately 15k training instances per scenario, consistent with the data scale in prior work (Zhang et al., 2025b). Both components are trained on the same set of open-source LLMs, such as Qwen2.5-3B, using full-parameter fine-tuning with a learning rate of 1e-5. The complete set of prompts used in our experiments is provided in the App. E. All experiments are conducted on a server equipped with eight NVIDIA A800-SXM4-80GB GPUs.

### 3.2.2 METRIC DESIGN

We evaluate translated code along two dimensions: **Computational Accuracy (CA)** and **Execution Time (ET)**. Computational Accuracy measures the proportion of translated programs that produce outputs identical to the source code across all inputs, following the standard metric used in prior work (Zhang et al., 2025b). Execution Time is defined as the average runtime of the translated code over all program inputs. For functionally incorrect translations, we use the baseline execution time from the benchmark as their runtime. To ensure reliable evaluation, we employ the Judge0 engine (Došilović & Mekterović, 2020), an online sandbox widely used for program execution testing (Waghjale et al., 2024). Each program, together with its inputs, is submitted to Judge0 and executed five times. The average runtime is then reported as the final result.

### 3.2.3 BASELINES

Our experiments include both training-free and training-based baselines. For the training-free baselines, we evaluate three prompt learning strategies on Qwen3-Next-80B (Qwen, 2025) and GPT-5: (1) **Correctness-Only**: prompts focusing solely on functional correctness. (2) **Correctness+Efficiency**: prompts emphasizing both correctness and runtime efficiency. (3) **Correctness→Efficiency**: a two-step prompting approach where the first step generates a correctness-oriented translation, which is then further optimized for efficiency. Detailed prompts for these training-free baselines are listed in the App. E. For the training-based baseline, we adopt **F2STrans** (Zhang et al., 2025b), which first applies IFT on weakly supervised data, followed by preference learning with high-quality data.

### 3.3 MAIN RESULTS

We implement our SWIFTTRANS framework based on Qwen2.5-3B, Qwen2.5-7B, Deepseek-6.7B and StarCoder-7B separately. Tab. 1 summarizes results on CodeNet, F2SBench, and SWIFTBENCH across five programming languages (C, C++, Go, Java, Python), reporting averages from each source language to the other four targets. App. C presents additional benchmark results, including PIE (Shypula et al., 2024) and xCodeEval (Khan et al., 2024).

**Functional Correctness Evaluation.** Tab. 1 (I) shows that prompts aimed at improving efficiency often significantly reduce functional correctness, even for GPT-5. This is intuitive, as introducing efficiency-oriented constraints increases the complexity of code translation, amplifying the risk of logical errors. Although more powerful LLMs such as GPT-5 are more robust to this trade-off, their high inference costs hinder wide application. In contrast, With our SWIFTTRANS framework, Qwen2.5-3B achieves an average CA 2.3% higher than F2STrans with Qwen2.5-7B, even though F2STrans leverages the stronger 7B model. Furthermore, applying SWIFTTRANS to Qwen2.5-7B outperforms GPT-5 by 3.8%. These results highlights both the potential of open-source LLMs for code translation and the effectiveness of SWIFTTRANS.

**Runtime Efficiency Evaluation.** From Tab. 1 (II), we can find that the "Correctness + Efficiency" and "Correctness→Efficiency" strategies do improve runtime efficiency, confirming that the target code translated by LLMs usually has significant room for efficiency improvement. However, these gains come at the expense of a decline in functional correctness, making these prompt engineering strategies suboptimal solutions. Moreover, scaling up F2STrans from 3B to 7B does not improve the runtime efficiency of translations. This stems from F2STrans's explicit emphasis on preserving the source code's logical structure (Zhang et al., 2025b), which mitigates errors but constrains runtime efficiency. In contrast, SWIFTTRANS employs multi-perspective exploration to generate diverse

Table 1: Functional correctness and runtime efficiency of translated code on the CodeNet, F2SBench, and SWIFTBENCH benchmarks. Each piece of data reflects the average performance for translations from one source language into the other four among C, C++, Go, Java, and Python.

| Method | LLM | CodeNet | | | | | F2SBench | | | | | SWIFTBENCH (Ours) | | | | | Avg. |
|---|---|---|---|---|---|---|---|---|---|---|---|---|---|---|---|---|---|
| | | C | C++ | Go | Java | Py | C | C++ | Go | Java | Py | C | C++ | Go | Java | Py | |
| **(I) Functional Correctness Evaluation—Computational Accuracy (%) ↑** | | | | | | | | | | | | | | | | | |
| Cor.-Only | Qwen3-Next-80B | 79.3 | 81.5 | 71.2 | 77.3 | 80.9 | 69.7 | 61.0 | 64.8 | 75.2 | 50.4 | 75.1 | 75.8 | 84.2 | 81.6 | 71.4 | 73.3 |
| Cor.+Eff. | | 79.7 | 79.2 | 66.8 | 74.6 | 77.9 | 66.0 | 51.7 | 55.1 | 68.3 | 44.6 | 73.5 | 77.6 | 78.4 | 70.7 | 65.5 | 68.6 |
| Cor.→Eff. | | 68.9 | 72.9 | 69.5 | 69.3 | 70.1 | 50.0 | 43.9 | 48.0 | 50.9 | 35.5 | 61.7 | 60.5 | 67.6 | 62.3 | 58.1 | 59.3 |
| Cor.-Only | GPT-5 | 87.8 | 91.4 | 91.9 | 81.8 | 90.3 | 88.0 | 81.4 | 85.6 | 88.1 | **63.8** | 90.0 | 82.3 | 92.8 | 91.1 | 90.4 | 86.4 |
| Cor.+Eff. | | 82.9 | 88.5 | 89.1 | 81.2 | 80.5 | 79.9 | 72.9 | 78.5 | 84.1 | 50.1 | 83.3 | 75.0 | 88.3 | 79.8 | 83.6 | 79.8 |
| Cor.→Eff. | | 68.4 | 62.9 | 66.9 | 70.2 | 61.3 | 62.3 | 46.3 | 52.3 | 58.4 | 49.2 | 74.9 | 57.1 | 67.9 | 78.5 | 63.3 | 62.7 |
| F2STrans [ICML 2025] | Qwen2.5-3B | 86.4 | 89.8 | 85.6 | 86.5 | 83.6 | 84.8 | 73.0 | 79.4 | 85.2 | 44.8 | 86.6 | 86.5 | 90.9 | 87.2 | 79.9 | 82.0 |
| | Qwen2.5-7B | 91.0 | 91.4 | 86.8 | 88.5 | 91.1 | 85.6 | 75.6 | 82.2 | 86.7 | 49.6 | 87.8 | 88.6 | 92.8 | 88.4 | 83.1 | 84.6 |
| **SWIFTTRANS (Ours)** | Qwen2.5-3B | 91.8 | 92.7 | 89.7 | 93.4 | 94.0 | 87.5 | 80.5 | 81.4 | 88.5 | 59.9 | 89.1 | 84.3 | 91.7 | 91.3 | 88.1 | 86.9 |
| | Qwen2.5-7B | 93.6 | 95.0 | 96.1 | 94.9 | 94.6 | 91.2 | 82.7 | 86.9 | 90.3 | 62.1 | 93.1 | 92.3 | 96.5 | 93.1 | 91.5 | 90.2 |
| | Deepseek-6.7B | 92.3 | 94.2 | 95.2 | 94.1 | 95.1 | 90.4 | 82.2 | 86.7 | 89.3 | 62.0 | 92.2 | 91.1 | 95.1 | 92.4 | 91.3 | 89.6 |
| | StarCoder-7B | 92.3 | 93.8 | 95.4 | 94.3 | 94.8 | 89.7 | 82.0 | 85.4 | 88.7 | 61.5 | 92.4 | 91.5 | 95.4 | 92.1 | 91.0 | 89.4 |
| **(II) Runtime Efficiency Evaluation—Execution Time (ms) ↓** | | | | | | | | | | | | | | | | | |
| Cor.-Only | Qwen3-Next-80B | 514 | 685 | 363 | 174 | 315 | 1397 | 1164 | 523 | 257 | 356 | 1651 | 1729 | 983 | 856 | 682 | 776 |
| Cor.+Eff. | | 455 | 529 | 295 | 173 | 256 | 1274 | 814 | 419 | 222 | 339 | 1509 | 1538 | 823 | 774 | 586 | 667 |
| Cor.→Eff. | | 364 | 504 | 285 | 121 | 228 | 936 | 769 | 395 | 222 | 284 | 1186 | 1211 | 782 | 656 | 542 | 565 |
| Cor.-Only | GPT-5 | 391 | 435 | 355 | 161 | 187 | 766 | 801 | 373 | 223 | 288 | 1010 | 1071 | 783 | 594 | 484 | 528 |
| Cor.+Eff. | | 376 | 357 | 338 | 137 | 167 | 690 | 721 | 309 | 172 | 257 | 870 | 880 | 623 | 512 | 388 | 453 |
| Cor.→Eff. | | 322 | 329 | 328 | 126 | 143 | 645 | 675 | 278 | **131** | **197** | 747 | 753 | 582 | 394 | 344 | 399 |
| F2STrans [ICML 2025] | Qwen2.5-3B | 494 | 613 | 340 | 164 | 336 | 1239 | 1006 | 440 | 270 | 522 | 1532 | 1694 | 985 | 897 | 638 | 744 |
| | Qwen2.5-7B | 470 | 711 | 303 | 175 | 320 | 1228 | 1089 | 423 | 261 | 508 | 1518 | 1599 | 837 | 884 | 639 | 731 |
| **SWIFTTRANS (Ours)** | Qwen2.5-3B | 218 | 269 | 223 | 138 | 146 | 561 | 593 | 252 | 218 | 239 | 609 | 686 | 384 | 322 | 238 | 339 |
| | Qwen2.5-7B | 190 | 216 | 145 | 106 | 122 | 472 | 573 | 203 | 168 | 217 | **563** | 551 | 328 | 313 | 214 | 292 |
| | Deepseek-6.7B | 192 | 226 | 151 | 107 | 125 | 475 | 580 | 212 | 168 | 221 | **563** | 558 | 329 | 317 | 217 | 296 |
| | StarCoder-7B | 203 | 242 | 168 | 123 | 126 | 497 | 577 | 228 | 186 | 233 | 579 | 605 | 352 | 316 | 232 | 311 |

translations, facilitating the selection of candidates that better balance correctness and efficiency. For example, the execution time of code translated by Qwen2.5-7B-based SWIFTTRANS is comparable to that of code produced by GPT-5 under the "Correctness→Efficiency" strategy.

## 3.4 ANALYSIS

We conduct detailed experiments to analyze our SWIFTTRANS framework. Unless otherwise specified, the experiments are based on SWIFTTRANS with Qwen2.5-3B and evaluated on the SWIFTBENCH benchmark. Further discussions on SWIFTTRANS are provided in App. D.

**Multi-Perspective Translation via Parallel ICL.** In the multi-perspective translation, each perspective is constructed with $k \in [0, 3]$ demonstrations, and we sample a total of $m = 10$ perspectives to generate 10 candidate translations. Fig. 3 illustrates the effect of $k$ and $m$ on performance. We evaluate $k$ under two settings: (i) fixed at 0, 1, 2, or 3, and (ii) randomly varying within [0,3].

We observe that while ICL brings substantial benefits within our SWIFTTRANS framework, the gains diminish as the number of demonstrations increases. For example, with $m = 10$ candidates, increasing $k$ from 0 to 1 improves CA by 1.3% and reduces ET by 142 ms, whereas increasing $k$ further from 1 to 3 provides only an additional 0.7% improvement in CA and 18 ms reduction in ET. Compared with using a fixed number of demonstrations, allowing $k$ to vary within [0, 3] delivers larger gains. This is because translations generated under variable-$k$ are

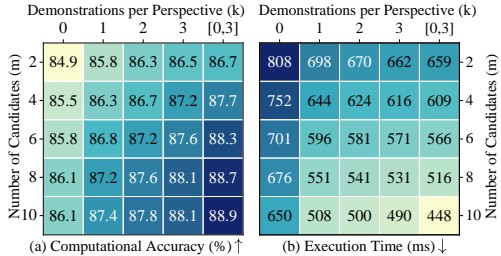

Figure 3: Effect of the number of demonstrations per perspective and the number of translation candidates in multi-perspective translation.

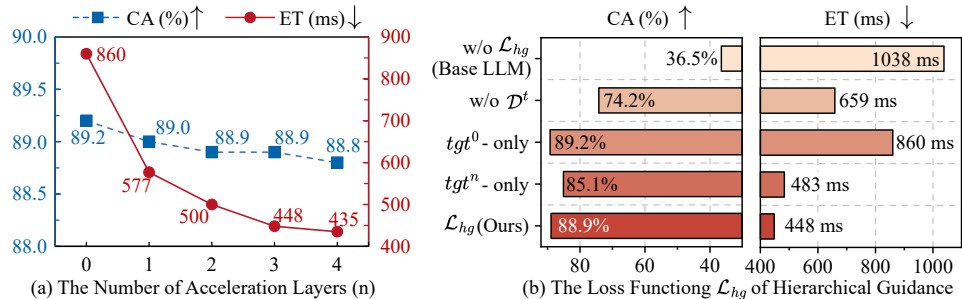

Figure 4: Analysis of the number of acceleration layers $n$ and the training loss function $\mathcal{L}_{hg}$ in hierarchical guidance.

essentially an aggregation of translations from multiple fixed-$k$ settings, leading to a more diverse candidate pool. In addition, increasing the number of candidates $m$ consistently improves translation quality. This corroborates prior findings (Brown et al., 2024) that multiple generations from the same prompt can help push the boundaries of LLM performance.

**MpTranslator Optimization via Hierarchical Guidance.** We analyze two key aspects of the hierarchical guidance strategy: the number of acceleration layers $n$ used in hierarchical data construction and the loss function $\mathcal{L}_{hg}$ in Eq. 1.

▶ *The Number of Acceleration Layers $n$.* The results of SWIFTTRANS applying various numbers of acceleration layers are shown in Fig. 4 (a). It can be observed that accelerating target code in the training data substantially mitigates efficiency issues in LLM-translated code, and additional acceleration layers further improve runtime efficiency. However, this comes at a slight cost to functional correctness—although the overall effect remains positive. For example, increasing $n$ from 0 to 4 significantly reduces ET by 425 ms, at the cost of a marginal decrease (0.4%) in CA.

▶ *The Loss Function $\mathcal{L}_{hg}$ of Hierarchical Guidance.* To analyze $\mathcal{L}_{hg}$, we define the following ablated variants of SWIFTTRANS: (1) "w/o $\mathcal{L}_{hg}$": candidate translations are generated directly by the base LLM, without hierarchical guidance training; (2) "w/o $\mathcal{D}^t$": demonstrations $\mathcal{D}^t$ are removed from $\mathcal{L}_{hg}$; (3) "$tgt^0$-only": only the correctness-first translation $tgt^0$ is used as the supervision signal; (4) "$tgt^n$-only": only the optimal translation $tgt^n$ is used as the supervision signal.

Fig. 4 (b) shows that hierarchical guidance substantially improves the code translation performance of base LLMs on both CA and ET. The sharp performance drop in the "w/o $\mathcal{D}^t$" variant highlights the importance of ICL-based training for maintaining consistency between training and inference. Neither the "$tgt^0$-only" nor the "$tgt^n$-only" variant achieves balanced performance: The former fails to promote runtime optimization (ET = 860 ms), while the latter over-prioritizes efficiency at the expense of correctness (CA = 85.1%). In contrast, SWIFTTRANS enables the model to maintain high correctness while improving efficiency.

**Bubble Selection via Difference-Aware Judge.** Inspired by bubble sort, we introduce a bubble selection strategy to accelerate the candidate selection process of DiffSelector. We compare bubble selection with all-pair selection, which evaluates all candidate pairs before selecting the best one. As shown in Tab. 2, bubble selection matches the quality of all-pair selection while reducing comparisons from $\mathcal{O}(n^2)$ to $\mathcal{O}(n)$. Specifically, all-pair selection outperforms bubble selection by just 0.2% in CA and 9 ms in ET. Given this

Table 2: Comparison between all-pair and bubble selection. All-pair selection judges every candidate pair before choosing the best, whereas bubble selection, inspired by bubble sort, significantly reduces the number of comparisons.

| Method | CA (%) ↑ | ET (ms) ↓ | # Judge ↓ |
|---|---|---|---|
| All-Pair. | **89.1** | **439** | $\mathcal{O}(n^2)$ |
| Bubble. | 88.9 | 448 | $\mathcal{O}(n)$ |

marginal performance difference and the significant reduction in the number of comparisons, bubble selection proves to be highly practical for efficient candidate evaluation.

**DiffSelector Optimization via Ordinal Guidance.** Ordinal guidance uses the loss $\mathcal{L}_{og}$ (Eq. 2) to compare translation quality bidirectionally. Our analytical experiments on ordinal guidance examine

three ablated variants of DiffSelector: (1) "w/o $\mathcal{L}_{og}$", which reflects the base LLM without training; (2) "w/o `diff`", which removes translation-difference information from the prompt and instead applies the standard pairwise judging strategy during training and inference; (3) "w/o Bi-Judge", which randomly selects one order for each translation pair during training. Additionally, since the pairwise judging strategy can be influenced by the order of translations in the prompt (Zheng et al., 2023), we introduce the Order Sensitivity (OS) metric to measure this effect across judge model variants. OS quantifies the proportion of inconsistent judgments when the order of two translations is reversed. Lower OS values indicate greater model robustness to input order.

Tab. 3 shows that all three ablated variants lead to performance degradation across CA, ET, and OS metrics, confirming the effectiveness of the complete ordinal guidance framework. Focusing on OS, we find that the base LLM exhibits high order sensitivity, with 64.2% of its judgments influenced by input order rather than translation quality, underscoring the limitations of off-the-shelf LLMs (Zheng et al., 2023). By explicitly incorporating `diff` information between translations and adopting the bi-judge training strategy, our ordinal guidance reduces this ratio to 6.4%.

Table 3: Ablation study on ordinal guidance for DiffSelector. The Order Sensitivity (OS) metric measures how sensitive the judge model is to the input order of translation pairs.

| Method | CA (%) ↑ | ET (ms) ↓ | OS (%) ↓ |
|---|---|---|---|
| **SWIFTTRANS** | **88.9** | **448** | **6.4** |
| w/o $\mathcal{L}_{og}$ | 86.1 | 609 | 64.2 |
| w/o `diff` | 87.3 | 519 | 27.5 |
| w/o Bi-Judge | 87.7 | 497 | 18.7 |

Importantly, the `diff` information contributes more than the bi-judge strategy, indicating that explicit difference information is crucial for distinguishing between highly similar translations.

## 3.5 DISCUSSION

**A More Comprehensive Evaluation of Translated Code Quality.** Beyond functional correctness and runtime efficiency, Table 4 also evaluates the translated code on **Memory Usage** and **Cyclomatic Complexity**, the latter being a standard indicator of code maintainability. Although SWIFTTRANS is designed primarily to improve correctness and runtime performance, it also produces code that uses less memory and has lower cyclomatic complexity. This benefit arises because many of the optimizations it performs—such as leveraging library utilities or removing redundant logic—naturally simplify control flow and reduce memory consumption. Table 10 in the appendix summarizes the optimization types, with more than half contributing to improvements in these two metrics.

Table 4: Average memory usage and cyclomatic complexity of translated programs. Both F2STrans and our SWIFTTRANS use Qwen2.5-7B as the backbone.

| Method | Memory Usage (MB) ↓ | Cyclomatic Complexity ↓ |
|---|---|---|
| Qwen3-Next-80B | 27.6 | 6.5 |
| GPT-5 | 26.1 | 5.9 |
| F2STrans | 29.1 | 7.0 |
| **SWIFTTRANS** | **23.9** | **5.7** |

**Inference Efficiency of the SWIFTTRANS Framework.** Although generating multiple candidates and running the judge introduces additional inference cost, we highlight two points: (1) under the same inference budget, SWIFTTRANS still outperforms F2STrans, and (2) candidate generation in SWIFTTRANS is fully parallelizable, so the extra overhead remains limited.

Table 5 reports functional correctness and average per-sample inference time for various translation frameworks. Due to their larger model sizes, Qwen3-Next-80B and GPT-5 incur much higher latency than F2STrans and SWIFTTRANS. With a single generated candidate, SWIFTTRANS has nearly the same inference time as F2STrans (a

Table 5: Functional correctness and average inference time per sample across various code translation frameworks.

| Method | Functional Correctness (%) ↑ | Inference Time (s) ↓ |
|---|---|---|
| Qwen3-Next-80B | 73.3 | 21.8 |
| GPT-5 | 86.4 | 121.3 |
| F2STrans | 84.6 | **5.1** |
| **SWIFTTRANS** | | |
| w/ 1-candidate | 87.1 | 5.3 |
| w/ 5-candidate | 89.3 | 8.1 |
| w/ 10-candidate | **90.2** | 10.2 |

difference of only 0.2s) while achieving 2.5% higher correctness. Increasing the number of candidates from 1 to 10 roughly doubles the inference time but yields a 3.5% improvement in correctness.

**Evaluation on Additional Benchmarks.** While CodeNet, F2SBench, and SWIFTBENCH contain source programs from online programming platforms, we further evaluate SWIFTTRANS on benchmarks covering broader scenarios. These include the class-level ClassEval-T benchmark (Xue et al., 2025) and the repository-level AlphaTrans (Ibrahimzada et al., 2025a) and RepoTrans (Wang et al., 2024a) benchmarks. As shown in Table 6, SWIFTTRANS

Table 6: Functional correctness evaluation on real-world class-level and repository-level benchmarks.

| Method | Class level | Repository level | |
|---|---|---|---|
| | ClassEval-T | AlphaTrans | RepoTrans |
| Qwen3-Next-80B | 18.6 | 23.1 | 3.0 |
| GPT-4o | 25.7 | **29.1** | 4.0 |
| F2STrans | 21.6 | 16.6 | 0.0 |
| **SWIFTTRANS** | **28.4** | 27.5 | **7.3** |

maintains strong performance across these benchmarks. The only exception is AlphaTrans, where GPT-4o achieves 1.6% higher functional correctness. We attribute this to the fact that source programs in AlphaTrans are very long, averaging over 5,000 tokens, a setting in which GPT-4o has a clear advantage over Qwen2.5-7B.

## 4 RELATED WORK

A number of studies have investigated how to improve the functional correctness of code generated by LLMs. These efforts can be broadly divided into two categories: training-free and training-based methods. Classic prompt learning strategies, such as RAG (Bhattarai et al., 2024a;b), fall under training-free methods and have proven effective. Some studies leveraged compiler feedback to detect translation errors and guide LLM-based fixes (Yang et al., 2024; Pan et al., 2024; Ibrahimzada et al., 2025b). In contrast, training-based approaches employ well-designed training processes, which enable lightweight open-source LLMs to achieve translation performance comparable to proprietary models. For example, He et al. (2025) incorporated executability signals into training, substantially enhancing the executability of code. Zhang et al. (2025b) proposed a two-stage approach: IFT on weakly aligned data, followed by preference learning on high-quality contrastive data.

In addition to functional correctness, runtime efficiency is an important criterion for evaluating code quality (ISO/IEC25010, 2011). In the task of code generation, Gee et al. (2024) trained LLMs to produce efficient solutions to programming problems, thereby achieving end-to-end code generation with improved efficiency. Accelerating generated code via post-processing is another mainstream approach. For example, Shypula et al. (2024) investigated LLM-based strategies code acceleration using techniques such as RAG, CoT, and IFT. Zhang et al. (2025c) further enhanced LLMs' optimization capabilities through curriculum learning. Although runtime efficiency has been increasingly recognized as an important metric for evaluating code generation models (Huang et al., 2024), to the best of our knowledge, existing research on code translation still focuses primarily on functional correctness. We argue that ensuring both functional correctness and runtime efficiency in translated code is crucial for applying code translation LLMs in practical software development.

## 5 CONCLUSION

In this work, we proposed SWIFTTRANS, a novel code translation framework that ensures both functional correctness and runtime efficiency of translated programs. Given source code, SWIFTTRANS first uses MpTranslator to generate diverse candidates through a multi-perspective translation strategy, and then employs DiffSelector to select the correct and most efficient candidate after comparison. In addition, we introduced hierarchical guidance for MpTranslator and ordinal guidance for DiffSelector to better adapt LLMs to these two core components. To support runtime efficiency evaluation, we extended functionality-oriented benchmarks (CodeNet, F2SBench) and constructed a new benchmark, SWIFTBENCH. Extensive experiments across these three benchmarks demonstrate that SWIFTTRANS significantly improves the quality of LLM-based code translation.

ETHICS STATEMENT

This work focuses on automated code translation and program optimization using large language models. It does not involve human subjects, personally identifiable information, or sensitive data. All datasets used—CodeNet, F2SBench, and our newly constructed SWIFTBENCH—are publicly available or derive from publicly accessible online programming platforms. We ensured that no proprietary or private codebases were included. The primary ethical consideration pertains to the deployment of automatically translated code in safety-critical or high-stakes systems. To mitigate such risks, we emphasize that our framework should be applied with human oversight and proper software validation. Our contributions are intended for academic research and general-purpose software engineering scenarios, and we do not foresee any directly attributable risks of security or privacy violations from this work.

REPRODUCIBILITY STATEMENT

We have taken extensive measures to ensure the reproducibility of our results. All benchmarks used in this work (CodeNet, F2SBench, and SWIFTBENCH) are publicly available or will be released upon acceptance of this paper. We provide full details of experimental settings, including training data construction, prompt templates, and evaluation metrics. Implementation details such as the number of demonstrations per perspective, the depth of hierarchical acceleration, and the loss functions used for optimization are described in Sec. 3.2.1. For runtime evaluation, we employed the Judge0 execution sandbox, a widely used open-source platform, as shown in Sec. 3.2.2. To further support reproducibility, we will release the source code for our SWIFTTRANS framework, including data processing scripts, training configurations, and evaluation pipelines. These materials will allow other researchers to replicate our experiments and validate our findings across different hardware setups.

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

APPENDIX

## A   THE USE OF LARGE LANGUAGE MODELS (LLMS)

During the preparation of this paper, LLMs were used as an auxiliary tool for language refinement and formatting. Specifically, GPT-based models were employed to enhance writing clarity, improve grammatical accuracy, and generate alternative phrasings for certain sentences. However, LLMs played no role in generating the research ideas, methodology, experimental design, or results. All conceptual contributions, technical developments, and data analyses were carried out by the authors. The final content was thoroughly verified and revised by the authors, who take full responsibility for the correctness and integrity of this work.

## B   BENCHMARK ANALYSIS

Table 7: Data statistics of CodeNet, F2SBench, and SWIFTBENCH.

| Benchmark | Language | #Code | #Cases | Date |
|---|---|---|---|---|
| CodeNet | C, C++, Go, Java, Python | $200 \times 5$ | 10 | Pre-2021 |
| F2SBench | C, C++, Go, Java, Python | $1000 \times 5$ | 10 | Mid-2024 |
| **SWIFTBENCH (Ours)** | C, C++, Go, Java, Python | $500 \times 5$ | 10 | Jun.–Aug. 2025 |

Table 8: Average execution time (ms) of conservative translations across benchmarks.

| Benchmark | {}$\to$ C | {}$\to$ C++ | {}$\to$ Go | {}$\to$ Java | {}$\to$ Python |
|---|---|---|---|---|---|
| CodeNet | 241 | 358 | 402 | 820 | 594 |
| F2SBench | 296 | 431 | 714 | 1486 | 1290 |
| **SWIFTBENCH (Ours)** | 718 | 578 | 801 | 1814 | 1400 |

Tab. 7 presents the data statistics for CodeNet, F2SBench, and SWIFTBENCH. Additionally, Tab. 8 illustrates the average execution time of annotated conservative translations on these three benchmarks. It can be observed that the code samples in CodeNet tend to be relatively simple. In contrast, the source code in SWIFTBENCH is intentionally designed to include efficiency issues, resulting in slower execution times for the translated code. This highlights the challenging nature of the SWIFTBENCH benchmark.

## C   ADDITIONAL RESULTS

We further evaluate SWIFTTRANS on the PIE (Shypula et al., 2024) and xCodeEval (Khan et al., 2024) benchmarks. Although PIE is commonly used for the code optimization task and xCodeEval for the code generation task, both provide source code along with basic test cases, making them suitable for evaluating code translation models. Notably, ET metric is not supported on these benchmarks, due to the lack of efficiency-critical test cases and the maximum baseline execution time derived from

Table 9: The code translation performance of various models on PIE (Shypula et al., 2024) and xCodeEval (Khan et al., 2024). Since these two benchmarks do not support evaluating the runtime efficiency of translated code, we report only functional correctness, *i.e.,* the CA metric.

| Method | LLM | PIE | xCodeEval | | | | | Avg. |
| | | C++ | C | C++ | Go | Java | Py | |
|---|---|---|---|---|---|---|---|---|
| Cor.-Only | Qwen3-Next | 63.6 | 78.7 | 67.8 | 82.5 | 70.6 | 69.6 | 72.1 |
| Cor.-Only | GPT-5 | **93.9** | 89.4 | 89.7 | 90.9 | 88.3 | 84.8 | 89.5 |
| F2STrans | Qwen2.5-3B | 86.4 | 90.4 | 87.0 | 90.3 | 87.2 | 82.2 | 87.2 |
| [ICML 2025] | Qwen2.5-7B | 89.2 | 91.3 | 88.8 | 91.8 | 89.4 | 84.1 | 89.1 |
| **SWIFTTRANS** | Qwen2.5-3B | 90.4 | 91.7 | 87.4 | 92.1 | 87.3 | 85.9 | 89.1 |
| **(Ours)** | Qwen2.5-7B | 92.3 | **93.1** | **90.6** | **93.5** | **92.4** | **89.6** | **91.9** |

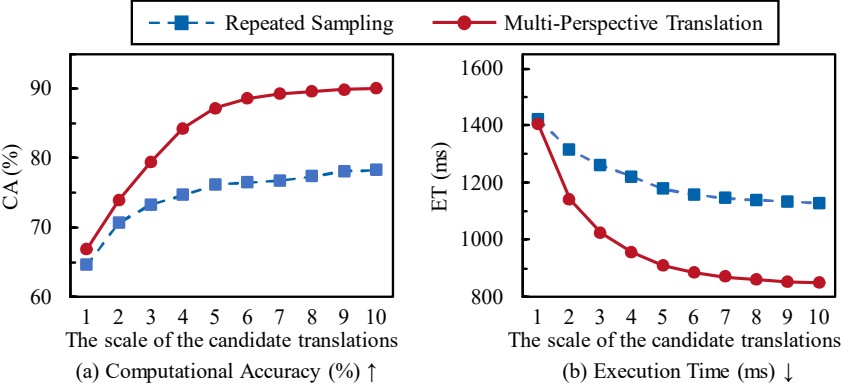

(a) Computational Accuracy (%) ↑      (b) Execution Time (ms) ↓

Figure 5: Comparison between the classic repeated sampling strategy and our multi-perspective translation strategy. In the experiment, Qwen3-Next-80B is used to generate multiple candidate Python translations for the C source code in the SWIFTBENCH benchmark, and the optimal one is selected.

conservative translations. Tab. 9 presents the performance of various models on these two benchmarks. We can find that the advantages of our SWIFTTRANS remain significant in both benchmarks. For example, the average CA of SWIFTTRANS based on Qwen2.5-7B exceeds that of GPT-5.

## D DISCUSSION

**Comparison between Repeated Sampling and Multi-Perspective Translation.** We directly compare the classic repeated sampling approach with our multi-perspective translation strategy. We apply both translation strategies using Qwen3-Next-80B to translate the C-to-Python subset of SWIFTBENCH benchmark. Fig. 5 shows the pass@k results, where the best candidate translation is selected directly, without any judging process. It is evident that multi-perspective translation brings larger gains than repeated sampling. For instance, under multi-perspective translation, pass@10 improves by 23.2% over pass@1 on the CA metric, whereas repeated sampling only gains 13.7%. Furthermore, at pass@10, multi-perspective translation significantly outperforms repeated sampling on both CA and ET. These results confirm that our multi-perspective translation provides higher-quality candidates than simple repeated sampling.

**Categorization of Efficiency-Oriented Translation Optimizations.** We classify code optimization patterns into six categories: Leveraging Language/Library Tools, Mathematical Simplification, Optimizing Algorithm Complexity, Removing Redundant Logic, Upgrading Data Structures, and Others. To estimate the prevalence of each type, we randomly sample 500 translations produced by Qwen2.5-3B-based SWIFTTRANS on SWIFTBENCH and compare them with manually annotated SWIFTBENCH translations that are correct but inefficient. If multiple categories were involved in one example, we selected the one with the greatest impact. As shown in Tab. 10, most optimizations

Table 10: Distribution of optimization categories in 500 randomly sampled translations from Qwen2.5-3B-based SWIFTTRANS on SWIFTBENCH.

| Optimization Category | Percentage |
|---|---|
| Leveraging Language/Library Tools | 20.1% |
| Mathematical Simplification | 6.4% |
| Optimizing Algorithm Complexity | 13.4% |
| Removing Redundant Logic | **30.5%** |
| Upgrading Data Structures | 26.4% |
| Others | 3.2% |

fall into three categories: Removing Redundant Logic, Upgrading Data Structures, and Leveraging Language/Library Tools.

# E    PROMPT SETTINGS

---

**Multi-Perspective Translation.**

Translate the following {SOURCE_LANG} code into {TARGET_LANG} code, maintaining functionality, and optimizing for performance:
### {SOURCE_LANG} Code:
{SOURCE_CODE}
### {TARGET_LANG} Code:

---

**Difference-Aware Judge.**

Here is a {SOURCE_LANG} code snippet and its translated {TARGET_LANG} version. Does my refinement to the {TARGET_LANG} code improve its correctness or efficiency?
### {SOURCE_LANG} Code:
{SOURCE_CODE}
### {TARGET_LANG} Code:
{TARGET_CODE_1}
### Refinement:
diff({TARGET_CODE_1}, {TARGET_CODE_2})

---

**Translation Layer of Hierarchical Data Construction.**

Translate the {SOURCE_LANG} code to {TARGET_LANG} code.
### {SOURCE_LANG} Code:
{SOURCE_CODE}
### {TARGET_LANG} Code:

---

**Acceleration Layer of Hierarchical Data Construction.**

Below is a {SOURCE_LANG} code. Optimize the code and provide a more efficient version.
### {SOURCE_LANG} Code:
{SOURCE_CODE}
### Optimized Version:

---

**Correctness-Only Prompt.**

Translate the {SOURCE_LANG} code to {TARGET_LANG} code.
### {SOURCE_LANG} Code:
{SOURCE_CODE}
### {TARGET_LANG} Code:

---

---

**Correctness+Efficiency Prompt.**

Translate the following {SOURCE_LANG} code into {TARGET_LANG} code, maintaining functionality, and optimizing for performance:
### {SOURCE_LANG} Code:
{SOURCE_CODE}
### {TARGET_LANG} Code:

---

**Correctness→Efficiency Prompt.**

*Stage 1—Correctness-Only Prompt:*
Translate the {SOURCE_LANG} code to {TARGET_LANG} code.
### {SOURCE_LANG} Code:
{SOURCE_CODE}
### {TARGET_LANG} Code:
*Stage 2—Code Acceleration Prompt:*
Below is a {TARGET_LANG} code. Optimize the code and provide a more efficient version.
### {TARGET_LANG} Code:
{TARGET_CODE}
### Optimized Version:

## F    CASE STUDY

Fig. 6 illustrates a case study of SWIFTTRANS, highlighting its advantages over traditional correctness-first models. Direct translation of the source code often carries over suboptimal logic from the original or overlooks optimizations specific to the target language. In contrast, SWIFTTRANS is designed to overcome these issues and produce translations that are both more efficient and more accurate.

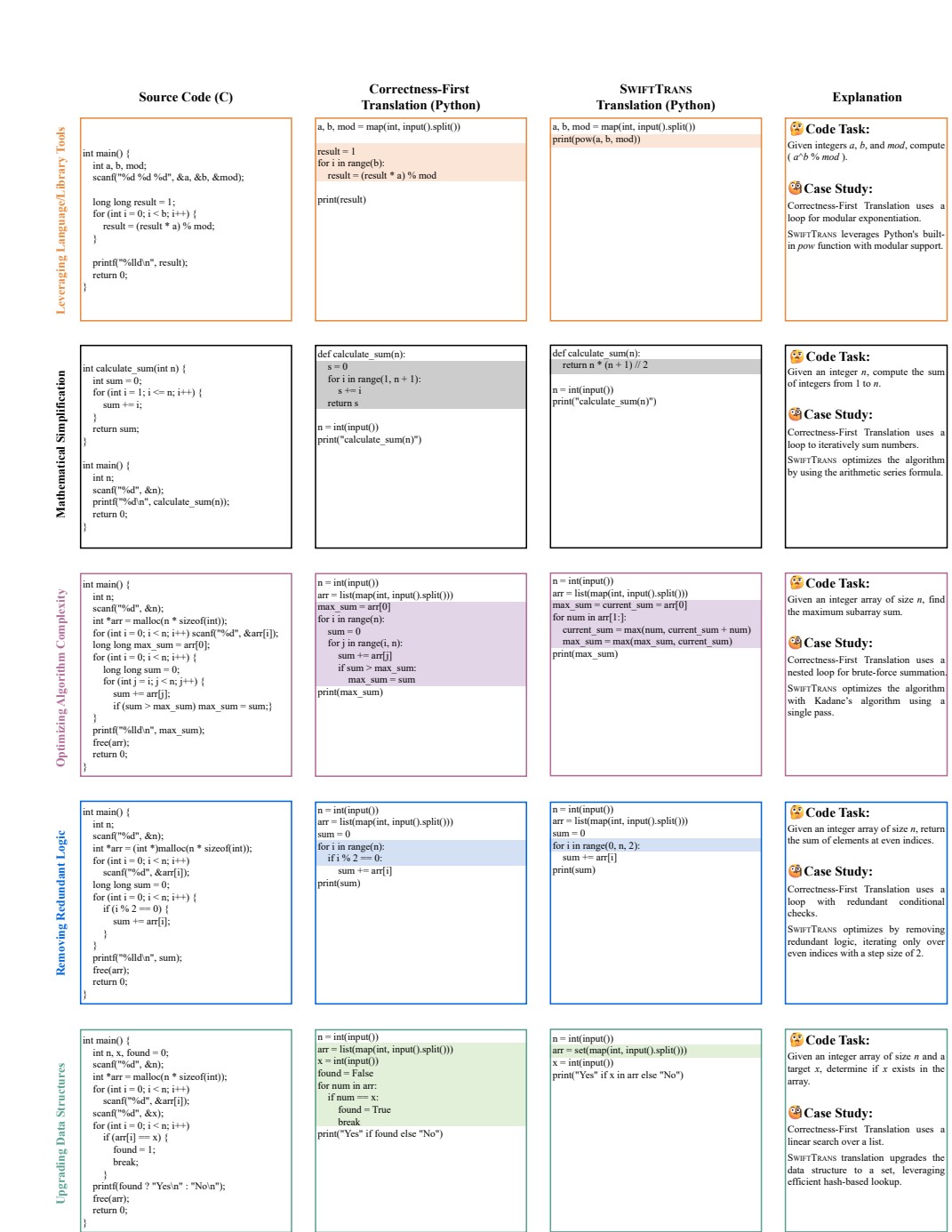

Figure 6: Case studies of SwiftTrans under different types of translation optimizations.

