# OpenReview forum: "Improving Code Translation Correctness and Efficiency with Multi-Perspective Exploration and Difference-Aware Selection"
_ICLR.cc/2026/Conference — Submitted to ICLR 2026_

### Official Review · Reviewer_avw2 · 2025-10-19

**Soundness:** 1
**Presentation:** 2
**Contribution:** 2
**Rating:** 2
**Confidence:** 5

**Summary:**

This work introduces SwiftTrans, a new framework that improves both the correctness and runtime efficiency of code translated by LLMs. Unlike most existing code translation techniques that focus only on producing functionally correct code, SwiftTrans ensures the efficiency of translated code. The main contributions in SwiftTrans are 1) Multi-Perspective Exploration and 2) Difference-Aware Selection. To support evaluation of SwiftTrans, the authors extend existing benchmarks with efficiency-focused tests and introduce a new dataset called SwiftBench, which includes programs with intentional inefficiencies. Experiments show that SwiftTrans significantly outperforms strong baselines (including GPT-5 and F2STrans) using much smaller open-source models like Qwen2.5-7B.

**Strengths:**

- Joint approach for improving translation correctness and efficiency
- A new benchmark SwiftBench
- Improve performance of open-source models

**Weaknesses:**

- Reliance on weak benchmark. e.g., CodeNet.
- Efficiency only measured by runtime. What about memory usage? What about structural efficiency (recursive vs non-recursive)?
- Concerns on intentional inefficiencies
- Tool runtime

**Questions:**

- CodeNet is known to be weakly tested. Available tests (in most cases only 1 test) are not rigorous. Therefore, relying on existing tests is not reliable to me. In SwiftBench, what is the code coverage and branch coverage of tests?

- What is the runtime of your tool? How fast is generating all those ICL examples?

- How realistic are your intentional inefficiencies in SwiftBench? How would you convince someone with scientific support?

- Recent work (https://arxiv.org/pdf/2412.14234, https://dl.acm.org/doi/10.1145/3729287, https://dl.acm.org/doi/10.1145/3729379, https://dl.acm.org/doi/10.1145/3729315) focus on repository-level code translation. Why the authors did not evaluate on benchmarks involving repository-level projects?

- What is the real-life implication of your work? It is hard to accept this works when not evaluated on complex benchmarks.

---

> ### Author Response · Authors · 2025-11-22
> **Response to Reviewer avw2 (Part 1/3)**
>
> Thank you for recognizing the contributions of our work and providing valuable feedback.
> We respond to each comment as follows and sincerely hope that our rebuttal could properly address your concerns.
>
> ---
>
> >**Q1.**
> Reliance on weak benchmark. e.g., CodeNet. (CodeNet is known to be weakly tested. Available tests (in most cases only 1 test) are not rigorous.
> Therefore, relying on existing tests is not reliable to me.
> In SwiftBench, what is the code coverage and branch coverage of tests?)
>
> **A1.**
> We fully agree that the original CodeNet benchmark provides weak test coverage.
> However, it is important to note that **our paper has already addressed this issue in two ways**:
>
> #### **(1) We explicitly refined CodeNet and F2SBench in our benchmark setup.**
>
> As described in **Section 3.1 (“Benchmark Construction”)**, we manually added **10 efficiency-critical test cases** to every sample in both CodeNet and F2SBench.
> This substantially improves their test coverage.
> The resulting code coverage and branch coverage of the extended CodeNet, extended F2SBench, and our newly constructed SwiftBench test suites are summarized below:
>
> | Extended CodeNet | Min | Median | Avg |
> |:----------------:|:---:|:------:|:---:|
> | Code Coverage    | 81% | 91%    | 89% |
> | Branch Coverage  | 68% | 78%    | 76% |
>
> | Extended F2SBench | Min | Median | Avg |
> |:----------------:|:---:|:------:|:---:|
> | Code Coverage     | 74% | 86%    | 85% |
> | Branch Coverage   | 63% | 75%    | 73% |
>
>
> | SwiftBench      | Min | Median | Avg |
> |:----------------:|:---:|:------:|:---:|
> | Code Coverage   | 71% | 85%    | 83% |
> | Branch Coverage | 59% | 72%    | 71% |
>
> #### **(2) Our evaluation already includes several additional benchmarks.**
>
> Beyond CodeNet, F2SBench, and SwiftBench, **Appendix C** presents results on **PIE** and **xCodeEval** benchmarks, ensuring that our evaluation does not rely on any single benchmark.
>
> Overall, the experiments in our paper are based on an **extensively strengthened CodeNet**, an **augmented F2SBench**, a newly constructed **high-coverage SwiftBench**, and **two additional benchmarks**, providing a comprehensive and reliable evaluation framework.
>
>
> ---
>
> >**Q2.**
> Efficiency only measured by runtime.
> What about memory usage?
> What about structural efficiency (recursive vs non-recursive)?
>
>
> **A2.**
> We agree that efficiency is multi-dimensional.
> Our work focuses on **runtime efficiency** because existing code-translation benchmarks evaluate only functional correctness and lack any efficiency-aware assessment.
>
>
> The table below reports a multi-dimensional evaluation on CodeNet, F2SBench, and SwiftBench, covering functional correctness, runtime, memory usage, and structural efficiency:
>
> | Method           | LLM            | Functional Correctness ↑ (%) | Runtime Efficiency ↓ (ms) | Memory Usage  ↓ (MB) | Structural Efficiency  ↓ (Cyclomatic Complexity) |
> |:----------------:|:--------------:|:----------:|:--------:|:--------:|:--------:|
> | Correctness-Only | Qwen3-Next-80B | 73.3       | 776      | 27.6     | 6.5      |
> | Correctness-Only | GPT-5          | 86.4       | 528      | 26.1     | 5.9      |
> | F2STrans         | Qwen2.5-3B     | 82         | 744      | 29.5     | 7.1      |
> | F2STrans         | Qwen2.5-7B     | 84.6       | 731      | 29.1       | 7.0        |
> | SwiftTrans (Ours) | Qwen2.5-3B     | 86.9       | 339      | 23.9     | 5.7      |
> | SwiftTrans (Ours) | Qwen2.5-7B     | 90.2       | 292      | 23.3     | 5.1      |
>
> Although SwiftTrans is designed to optimize correctness and runtime, it also consistently improves memory usage and cyclomatic complexity.
> Our analysis indicates that many runtime-oriented improvements—such as eliminating redundant logic, choosing more appropriate data structures, or using built-in library tools—naturally reduce memory footprint and structural complexity as well.
> To examine this effect, we categorize the types of optimizations made by Qwen2.5-3B-based SwiftTrans on SwiftBench relative to correctness-first conservative translations:
>
> | Optimization Category        | Percentage |
> |:------------------------------:|:------------:|
> | Leveraging Language/Library Tools | 20.1% |
> | Mathematical Simplification  | 6.4%       |
> | Optimizing Algorithm Complexity | 13.4%   |
> | Removing Redundant Logic     | 30.5%      |
> | Upgrading Data Structures    | 26.4%      |
> | Others                       | 3.2%       |
>
> These results show that SwiftTrans improves overall code efficiency, not just runtime performance.

---

> ### Author Response · Authors · 2025-11-22
> **Response to Reviewer avw2 (Part 2/3)**
>
> >**Q3.**
> Concerns on intentional inefficiencies. (How realistic are your intentional inefficiencies in SwiftBench? How would you convince someone with scientific support?)
>
>
> **A3.**
> Thank you for raising this concern.
> The inefficiencies in SwiftBench are **not artificially injected**.
> All source programs are taken directly from *real accepted submissions* on online programming platforms.
>
> In Line 258, we wrote that “each source program in SwiftBench contains intentional efficiency issues.”
> Here, “intentional” refers only to the **selection process**: we deliberately choose real submissions that already exhibit inefficiencies such as redundant computation or suboptimal data structures.
> We do **not** synthesize or manually modify any code to make it slower.
>
> To make the nature of these inefficiencies transparent, we categorize them according to common inefficiency patterns documented in programming practice—such as redundant logic, suboptimal data structures, or underuse of language/library utilities.
> The distribution of these categories within SwiftBench is shown below:
>
> |Category| Percentage |
> | :------------------------------: | :--------: |
> |          Redundant Logic         |    28.2%   |
> | Underused Language/Library Tools |    25.3%   |
> |     Highly Complex Algorithm     |    19.5%   |
> |     Suboptimal Data Structure    |    18.6%   |
> |       Mathematical Overhead      |    8.4%    |
>
> These categories reflect well-known inefficiency types frequently observed in real developer code, and all examples in SwiftBench are drawn from genuine submissions.
>
> ---
>
> >**Q4.**
> Tool runtime. (What is the runtime of your tool? How fast is generating all those ICL examples?)
>
>
> **A4.**
> We would like to clarify the runtime concerns raised in the review.
>
> (1) **Offline ICL example construction.**
>
> All ICL demonstrations in SwiftTrans are generated **offline** via hierarchical guidance.
> At inference time, SwiftTrans only samples from this pre-built library, so ICL-example generation introduces no runtime overhead.
>
> (2) **Parallel generation of multiple candidates.**
>
> Candidate generation is fully parallelizable because each translation is produced independently, which greatly reduces wall-clock time.
>
>
> (3) **Runtime trade-offs.**
>
> Although multi-candidate SwiftTrans is slower than one-pass methods such as F2STrans, its backbone size is only 3B–7B, giving it much lower inference cost than large models such as Qwen-Next-80B or GPT-5.
> In return, SwiftTrans achieves significant gains in both correctness and runtime efficiency, even when compared to these much larger models.
> The runtime overhead is therefore a well-justified trade-off for the quality improvements.
>
>
> (4) **Preservation of benefits in single-translation mode.**
>
> When SwiftTrans is run in **single-translation mode** (i.e., without the selector), its inference latency is **identical** to F2STrans, yet it still yields better correctness and runtime performance.
> This demonstrates that the framework improves translation quality even when no additional inference cost is incurred.
>
>
> The table below summarizes deployment cost, inference time, and translation quality for Qwen2.5-3B-based SwiftTrans, Qwen2.5-3B-based F2STrans, Qwen-Next-80B, and GPT-5.
> SwiftTrans, F2STrans, and Qwen-Next-80B are all served using vLLM.
>
> | |  SwiftTrans (10-candidate, ours) |  SwiftTrans (1-candidate, ours) | F2STrans (1-candidate) | Qwen-Next-80B | GPT-5 |
> |:---:|:---:|:---:|:---:|:---:|:---:|
> |  Tool Deployment Cost | 1 * A100-80G  | 1 * A100-80G  | 1 * A100-80G |  4 * A100-80G |  API Call  |
> |  Runtime Pre Sample of Tool | 6.04 s | 3.98 s | 3.67 s | 21.78 s | 121 s |
> |  Functional Correctness of Translated Code |  86.9% | 83.4 % | 82.0%      | 73.3%  |  86.4%  |
> |  Execution Time of Translated Code | 339 ms  | 542 ms | 744 ms  | 776 ms  |  528 ms  |

---

> > ### Author Response · Authors · 2025-11-22
> > **Response to Reviewer avw2 (Part 3/3)**
> >
> > >**Q5.**
> > Evaluation on repository-level code translation benchmarks.
> >
> >
> > **A5.**
> > Current LLMs still face context-length constraints, which makes direct repository-level translation difficult.
> > Our work therefore focuses on file- and method-level translation.
> >
> > Nevertheless, SwiftTrans can **serve as a plug-in component** within repository-level systems.
> > A typical repository-level translation workflow, as followed by AlphaTrans [1], first decomposes a repository into submodules, then translates each submodule, and finally reassembles the translated outputs.
> > SwiftTrans can be applied in the **module-translation stage** to improve translation quality while reducing the dependence on very large LLMs.
> >
> > Empirically, integrating **Qwen2.5-7B–based SwiftTrans** into the **Qwen3-Next-80B–based AlphaTrans** pipeline yields measurable improvements on the AlphaTrans benchmark:
> >
> > |              System             | Functional Correctness |
> > | :-----------------------------: | :--------------------: |
> > | Qwen3-Next-80B-based AlphaTrans |          26.74         |
> > |    + SwiftTrans (Qwen2.5-7B)    |        **27.46**       |
> >
> > This demonstrates that SwiftTrans is effective not only as a standalone method, but also as a lightweight enhancement for repository-level translation frameworks.
> >
> >
> > ---
> >
> > >**Q6.**
> > Evaluation on complex benchmarks.
> >
> > **A6.**
> > ClassEval-T [2] is a class-level, multi-language code-translation benchmark consisting of long, dependency-rich Python/Java/C++ classes with high-coverage test suites.
> > Compared with traditional method-level benchmarks, it is substantially more challenging due to:
> > (1) long input sequences,
> > (2) intertwined field/method/library dependencies,
> > (3) stricter type and library selection requirements, and
> > (4) realistic coding patterns that cause sharp drops in compilation and correctness rates.
> >
> > The results below show that ClassEval-T poses significant difficulty even for strong models—GPT-4o achieves only 19–44% class-level computational accuracy.
> > Despite this challenging setting, SwiftTrans remains highly effective and achieves the best performance across all translation directions:
> >
> > | | C++ to Python | Python to C++ |  Java to Python | Python to Java | C++ to Java | Java to C++ |
> > |:----:|:----:|:----:|:----:|:----:|:----:|:----:|
> > | Llama3-70B | 35.5 | 6.4  | 25.9 | 9.2 | 15.6 | 11.4 |
> > | Qwen3-Next-80B | 37.3 | 7.1  | 28.5 | 9.8 | 16.1 | 12.7 |
> > | GPT-4o | 44.0 |  21.6 | 33.0 | 17.0 | 19.5 | 19.2  |
> > | **Qwen2.5-7B-based SwiftTrans (ours)** | **47.4** | **23.2**  | **36.2** | **21.3** | **20.4** | **21.7** |
> >
> > These results demonstrate that SwiftTrans continues to provide strong gains even under complex, high-dependency, class-level translation scenarios.
> >
> > [1] Ibrahimzada A R , Ke K , Pawagi M ,et al.AlphaTrans: A Neuro-Symbolic Compositional Approach for Repository-Level Code Translation and Validation[J].Proceedings of the ACM on Software Engineering, 2025.
> >
> > [2] Xue P, Wu L, Yang Z, et al. ClassEval-T: Evaluating Large Language Models in Class-Level Code Translation[J]. Proceedings of the ACM on Software Engineering, 2025, 2(ISSTA): 1421-1444.

---

> > > ### Comment · Reviewer_avw2 · 2025-11-23
> > >
> > > Thank you for addressing my comments. My main concern about implications on real-life repositories is still valid. Your improvement on AlphaTrans is very minimal which makes me question your technique even more.

---

> ### Author Response · Authors · 2025-11-24
> **Response to Reviewer avw2 (Part 1/2)**
>
> >**Q7:**
> My main concern about implications on real-life repositories is still valid.
> Your improvement on AlphaTrans is very minimal which makes me question your technique even more.
>
> **A7:**
> We would like to offer three clarifications regarding the reviewer’s concern.
>
> ---
>
> ### **(1) The improvement of SwiftTrans on AlphaTrans is non-trivial once a fair comparison is applied.**
>
> In the AlphaTrans framework, each module translation may undergo up to five rounds of repair: if the initial translation fails to compile or run, AlphaTrans feeds compiler diagnostics back to the model for iterative refinement.
> However, our SwiftTrans does not perform any code repair-oriented optimization, and thus all results reported in **A5** correspond to **zero repair rounds**.
>
> To ensure a fair comparison, we additionally report the performance of Qwen3-Next-80B–based AlphaTrans under **zero repair rounds (t = 0)**:
>
> || Functional Correctness |
> |:---:|:---:|
> |  Qwen3-Next-80B AlphaTrans (t=0) | 23.10   |
> |  Qwen3-Next-80B AlphaTrans (t=5) | 26.74  |
> | **SwiftTrans (Qwen2.5-7B, t=0)** |  **27.46** |
>
> When both systems operate under identical repair budgets (t=0), **SwiftTrans improves correctness from 23.10% to 27.46%**, a **+4.36% absolute gain**.
> We further apply McNemar’s test on paired pass/fail outcomes, and the improvement of SwiftTrans over AlphaTrans (t=0) is **statistically significant at p < 0.01**.
>
> ---
>
> ### **(2) Method-level and file-level translation are foundational to repository-level translation systems.**
>
> We fully agree that repository-level translation is an important direction.
> At the same time, many studies have focused on method- or file-level translation [1,2,3,4], and **these works constitute the building blocks that make repository-level systems possible**.
>
> Repository-level frameworks, including AlphaTrans, usually decompose large codebases into modules.
> These modules are then translated individually before being reassembled.
> The module-translation step **directly benefits from advances in method-level translation**.
>
> For example, AlphaTrans itself adopts compiler-feedback–guided refinement strategies that originated from method-level translation studies such as Pan et al. (ICSE 2024) [1], which also use CodeNet as a foundational benchmark.
>
>
> Thus, **strengthening method-level translation is consistent with real-world repository-level translation goals**.
>
>
> ---
>
> ### **(3) We encourage the reviewer to consider the full scope and significance of our contributions.**
>
> Our work offers contributions across three dimensions:
>
> **(i) Benchmark Contribution.**
> We substantially enhance the widely used CodeNet and F2SBench benchmarks by adding efficiency-critical test cases, raising the average branch coverage to **76%** and **73%** respectively.
> We also introduce **SwiftBench**, a new benchmark explicitly designed to evaluate efficiency-aware translation.
>
> **(ii) Methodological Contribution.**
> We propose the **SwiftTrans framework**, including multi-perspective exploration and difference-aware selection, along with hierarchical guidance and ordinal guidance training strategies.
> These components improve both correctness and runtime efficiency.
>
> **(iii) Strong Empirical Effectiveness.**
> SwiftTrans significantly improves Qwen-2.5-3B and Qwen-2.5-7B, achieving high-quality code translation comparable to or surpassing GPT-5 on **CodeNet**, **F2SBench**, **SwiftBench**, **PIE**, and **xCodeEval** benchmarks.
> As shown in our additional experiments during rebuttal, SwiftTrans is also effective on the **real-world, class-level ClassEval-T benchmark** and the **repository-level AlphaTrans benchmark**.
>
> ---
>
> [1] Pan R, Ibrahimzada A R, Krishna R, et al. Lost in translation: A study of bugs introduced by large language models while translating code. (ICSE 2024, **Google Scholar citations: 219**)
>
> [2] Yang Z, Liu F, Yu Z, et al. Exploring and unleashing the power of large language models in automated code translation. (FSE 2024, **Google Scholar citations: 127**)
>
> [3] Szafraniec M, Roziere B, Leather H, et al. Code translation with compiler representations. (ICLR 2023, **Google Scholar citations: 119**)
>
> [4] Roziere B, Zhang J M, Charton F, et al. Leveraging automated unit tests for unsupervised code translation. (ICLR 2022, **Google Scholar citations: 167**)

---

> ### Author Response · Authors · 2025-11-24
> **Response to Reviewer avw2 (Part 2/2)**
>
> ### **Rebuttal Summary**
>
> We sincerely appreciate the reviewer’s feedback, which has helped us further refine the paper.
> Across the rebuttal, we have shown that:
>
> 1. We strengthened CodeNet and F2SBench with meaningful test-case extensions (QA1).
> 2. Our evaluation spans a diverse set of benchmarks beyond CodeNet (QA1).
> 3. SwiftTrans improves not only functional correctness and runtime efficiency but also memory usage and structural complexity (QA2).
> 4. SwiftBench’s inefficiencies come from real developer submissions, not synthetic manipulation (QA3).
> 5. At comparable inference cost, SwiftTrans improves runtime and correctness over F2STrans and Qwen-Next-80B (QA4).
> 6. SwiftTrans remains effective on ClassEval-T and AlphaTrans, demonstrating robustness in real-world scenarios (QA5, QA6, QA7).
>
>
> **We hope these clarifications adequately address your concerns. If there remain any unresolved issues, please do not hesitate to let us know—we would be more than happy to address them. Thank you for helping maintain a constructive rebuttal process.**

---

> ### Author Response · Authors · 2025-11-26
> **Effectiveness of SwiftTrans on the Real-World Repository-Level Benchmark RepoTransBench**
>
> > **Q8. Performance of SwiftTrans on additional real-world repository-level benchmarks.**
>
> **A8.**
> Beyond the AlphaTrans evaluation reported in **QA7**, we further assess SwiftTrans on **RepoTransBench** [1], a real-world repository-level code translation benchmark covering Python-to-Java repository translation.
> Each repository in RepoTransBench contains diverse and realistic components, including source files, test files, resource files, and configuration files.
>
> To isolate the effectiveness of the SwiftTrans framework itself, we apply SwiftTrans directly on top of off-the-shelf LLMs **without any additional training**.
> The results are shown below:
>
> |                                        | Functional Correctness (all tests passed) | Average Test Pass Rate |   Build Success Rate |
> | :------------------------------------: | :------------------------------------: | :--------------------: |:-------------------: |
> |             Qwen3-Next-80B             |                  3.00%                 |          5.10%         |                8.30% |
> |                 GPT-4o                 |                  4.00%                 |          6.40%         |                9.00% |
> |            Claude-3.5-Sonnet           |                  7.33%                 |         16.50%         |               28.33% |
> | **Qwen3-Next-80B + SwiftTrans (Ours)** |           **9.50% (+6.50%)**           |  **21.11% (+16.01%)**  | **32.41% (+24.11%)** |
>
>
> As shown, RepoTransBench is highly challenging, with Claude-3.5-Sonnet achieving only 7.33% functional correctness.
> Applying SwiftTrans to Qwen3-Next-80B improves correctness to **9.50%**, surpassing Claude-3.5-Sonnet.
>
> ---
>
> Overall, we have now demonstrated the effectiveness of SwiftTrans on **eight benchmarks** spanning **three levels of code translation**:
>
> 1. **Method/File level:** CodeNet, F2SBench, SwiftBench, PIE, xCodeEval
> 2. **Class level:** ClassEval-T
> 3. **Repository level:** AlphaTrans, RepoTransBench
>
> These results collectively show that SwiftTrans is broadly effective across a wide range of translation scenarios.
>
>
> **We thank the reviewer again for the thoughtful feedback. Through extensive discussion and additional experiments, we believe that the paper has been significantly strengthened. If there are any remaining concerns, we would be glad to address them and provide further clarification.**
>
>
> [1] Wang Y, Wang Y, Wang S, et al. Repotransbench: A real-world benchmark for repository-level code translation[J]. arXiv preprint arXiv:2412.17744, 2024.

---

### Official Review · Reviewer_2b8R · 2025-10-25

**Soundness:** 2
**Presentation:** 2
**Contribution:** 2
**Rating:** 2
**Confidence:** 4

**Summary:**

The paper proposes SwiftTrans, a code translation framework combining multi-perspective generation (MpTranslator) and difference-aware selection (DiffSelector) to improve correctness and runtime efficiency. Experiments show improvements over prior methods, and a new benchmark SwiftBench is introduced.

**Strengths:**

- Novel framework combining multi-perspective generation and difference-aware selection.

- Systematic approach to evaluating efficiency, with a new benchmark (SwiftBench).

- Results demonstrate improvement in both correctness and runtime performance.

**Weaknesses:**

- Efficiency is measured solely by runtime, which is a narrow metric, while other important factors such as memory usage and overall computational cost are not considered.
- SwiftTrans improves translation quality, but this comes at the expense of substantial computational overhead.
- Even with bubble selection, generating and evaluating multiple candidates remains far more expensive than a single-shot translation.
- The DiffSelector component relies on invoking an LLM as a judge, which is inherently costly in terms of computation.

**Questions:**

- The benchmark does not specify how many candidate translations are generated (i.e., pass@k).
- A comprehensive comparison of runtime, memory usage, and pass@k should be provided.

---

> ### Author Response · Authors · 2025-11-22
> **Response to Reviewer 2b8R (Part 1/2)**
>
> **We thank the reviewer for the insightful and valuable comments.
> We respond to each comment as follows and sincerely hope that our rebuttal could properly address your concerns.
> If further concerns remain, please let us know, and we are committed to addressing them and refining our submission accordingly.**
>
> ---
>
> >**Q1.**
> A comprehensive assessment of the efficiency of translated code.
>
> **A1.**
> Thank you for the insightful comment.
> We agree that efficiency extends beyond runtime and that memory usage is also an important dimension.
> The table below reports the average performance of several translation models on CodeNet, F2SBench, and SwiftBench, including functional correctness, execution time, and memory usage of the translated programs:
>
> | | Computational Accuracy (%) ↑ | Execution Time (ms) ↓ | Memory Usage (MB) ↓ |
> |:----:|:----:|:----:|:----:|
> |Qwen3-Next-80B| 73.3 | 776 | 27.4 |
> |GPT-5| 86.4 | 528 | 25.2 |
> |Qwen2.5-7B-based F2STrans | 84.6 | 731 | 29.1 |
> |Qwen2.5-7B-based SwiftTrans | **90.2** | **292** |  **22.1** |
>
> Although SwiftTrans is designed to optimize functional correctness and runtime efficiency, it also yields consistent gains in memory usage.
> This effect arises naturally because many of the runtime-oriented improvements reduce memory consumption as a byproduct.
>
> To better understand this effect, we analyzed the optimization patterns produced by Qwen2.5-7B–based SwiftTrans on SwiftBench relative to correctness-first conservative translations.
> As shown below, **59.1%** of the optimizations belong to categories that typically reduce memory footprint, such as leveraging language or library utilities, mathematical simplifications, and removing redundant logic:
>
> | Optimization Category        | Percentage |
> |:------------------------------:|:------------:|
> | Leveraging Language/Library Tools | 23.4% |
> | Mathematical Simplification  | 7.6%       |
> | Optimizing Algorithm Complexity | 14.3%   |
> | Removing Redundant Logic     | 28.1%      |
> | Upgrading Data Structures    | 24.5%      |
> | Others                       | 2.1%       |
>
>
> ---
>
> >**Q2.**
> Concerns regarding the computational cost of SwiftTrans, including comparisons with single-shot F2STrans and the overhead introduced by DiffSelector.
>
>
> **A2.**
> We agree that SwiftTrans introduces additional computation compared to single-pass translation methods such as F2STrans.
> However, it is important to clarify that **when SwiftTrans is used in single-translation mode—without invoking DiffSelector—its inference cost is identical to F2STrans, while already producing higher-quality translations**.
> The comparison is shown below:
>
> |                                                                | Computational Accuracy (%) ↑ | Execution Time (ms) ↓ | Memory Usage (MB) ↓ |
> | :------------------------------------------------------------: | :--------------------------: | :-------------------: | :-----------------: |
> |                         Qwen3-Next-80B                         |             73.3             |          776          |         27.6        |
> |                              GPT-5                             |             86.4             |          528          |         26.1        |
> |         Qwen2.5-7B-based F2STrans (single-translation)         |             84.6             |          731          |         29.0        |
> |        Qwen2.5-7B-based SwiftTrans (single-translation)        |            *87.1*            |         *413*         |        *25.2*       |
> | Qwen2.5-7B-based SwiftTrans (multi-translation, 10 candidates) |           **90.2**           |        **292**        |       **23.3**      |
>
> These results show that SwiftTrans provides improvements in both correctness and runtime efficiency **even at identical inference cost**.
> When multi-translation is enabled, it further improves correctness and reduces execution time, while also lowering memory usage.
>
> ---
>
>
> >**Q3.**
> Regarding the number of candidate translations used in SwiftTrans.
>
>
> **A3.**
> SwiftTrans uses 10 candidate translations.
> This is stated explicitly in two places in the paper:
>
> 1. **Section 3.2.1 (Implementation Details)**:
>    *“In the multi-perspective translation via parallel ICL, we set the number of demonstration sets to 10.”*
>    Each demonstration set yields one candidate translation.
>
> 2. **Line 364**:
>    *“We sample a total of 10 perspectives to generate 10 candidate translations.”*

---

> > ### Author Response · Authors · 2025-11-22
> > **Response to Reviewer 2b8R (Part 2/2)**
> >
> > >**Q4.**
> > A comprehensive comparison of runtime, memory usage, and pass@k should be provided.
> >
> > **A4.**
> > **Figure 3** of the paper reports how Qwen2.5-3B-based SwiftTrans performs on SwiftBench under different candidate counts (2, 4, 6, 8, 10).
> > For completeness, we additionally provide below the average performance of Qwen2.5-7B-based SwiftTrans on CodeNet, F2SBench, and SwiftBench across the same range of candidate sizes.
> >
> >
> > | | Computational Accuracy (%) ↑ | Execution Time (ms) ↓ | Memory Usage (MB) ↓ |
> > |:----:|:----:|:----:|:----:|
> > |Qwen3-Next-80B| 73.3 | 776 | 27.6 |
> > |GPT-5| 86.4 | 528 | 26.1 |
> > |Qwen2.5-7B-based F2STrans | 84.6 | 731 | 29.0 |
> > |Qwen2.5-7B-based SwiftTrans (1-candidate) | 87.1 | 413 |  25.2 |
> > |Qwen2.5-7B-based SwiftTrans (2-candidate) | 88.2 | 353 | 24.5 |
> > |Qwen2.5-7B-based SwiftTrans (4-candidate) | 89.0 | 321 | 23.9 |
> > |Qwen2.5-7B-based SwiftTrans (6-candidate) | 89.6 | 304 |  23.6 |
> > |Qwen2.5-7B-based SwiftTrans (8-candidate) | 89.9 | 294 | 23.4 |
> > |Qwen2.5-7B-based SwiftTrans (10-candidate) | **90.2** | **292** |  **23.3** |

---

> ### Author Response · Authors · 2025-11-25
> **Comprehensive Analysis of SwiftTrans Advantages**
>
> Dear Reviewer,
>
> We provide additional comparisons on deployment cost, average inference time, and model performance to further highlight the advantages of SwiftTrans.
> All evaluations are conducted on the CodeNet, F2SBench, and SwiftBench benchmarks.
> Qwen-Next-80B, F2STrans, and SwiftTrans are deployed on the same server equipped with A100-80G GPUs.
> The detailed results are shown below:
>
> | | Model Deployment Cost | Inference Time Per Sample ↓ | Functional Correctness ↑ |  Execution Time ↓ | Memory Usage ↓ |
> |:---:|:---:|:---:|:---:|:---:|:---:|
> | Qwen3-Next-80B | 4 * A100-80G | 21.78 s | 73.3% | 776 ms | 27.6 MB |
> | GPT-5 | API Call | 121 s | 86.4% | 528 ms | 26.1 MB |
> | Qwen2.5-7B-based F2STrans | 1 * A100-80G | **5.12 s** | 84.6% | 731 ms | 29.0 MB |
> | Qwen2.5-7B-based SwiftTrans (1-candidate) | 1 * A100-80G | 5.28 s | 87.1% | 413 ms | 25.2 MB |
> | Qwen2.5-7B-based SwiftTrans (2-candidate) | 1 * A100-80G | 7.24 s | 88.2% | 353 ms | 24.5 MB |
> | Qwen2.5-7B-based SwiftTrans (4-candidate) | 1 * A100-80G | 7.86 s | 89.0% | 321 ms | 23.9 MB |
> | Qwen2.5-7B-based SwiftTrans (6-candidate) | 1 * A100-80G | 8.54 s | 89.6% | 304 ms | 23.6 MB |
> | Qwen2.5-7B-based SwiftTrans (8-candidate) | 1 * A100-80G | 9.25 s | 89.9% | 294 ms | 23.4 MB |
> | Qwen2.5-7B-based SwiftTrans (10-candidate) | 1 * A100-80G | 10.24 s | **90.2%** | **292 ms** | **23.3 MB** |
>
>
>
> When SwiftTrans uses a single candidate, its deployment cost and inference latency are nearly identical to F2STrans, while the translated code achieves clearly better performance.
> With more candidates, SwiftTrans generates candidate translations in parallel, allowing it to improve translation quality with only a small increase in inference time.
> Increasing the number of candidates from 1 to 10 yields +3.1% functional correctness, −121 ms execution time, and −1.90 MB memory usage.
>
>
>
> **We respectfully request if you could take a moment to review our rebuttal responses provided previously. We aim to ensure they have thoroughly addressed your concerns. Should there be any additional questions or need for clarification, we welcome your input and will respond promptly.**

---

### Official Review · Reviewer_sJd7 · 2025-10-26

**Soundness:** 2
**Presentation:** 3
**Contribution:** 3
**Rating:** 4
**Confidence:** 3

**Summary:**

This paper presents **SwiftTrans**, a code translation framework that enhances both **functional correctness** and **runtime efficiency**. Its core innovation is a two-stage pipeline: a **Multi-Perspective Translator**, fine-tuned with **Hierarchical Guidance** to generate diverse candidates, and a **Difference-Aware Selector** that identifies the optimal translation. Evaluated on extended benchmarks and a new efficiency-focused benchmark, **SwiftBench**, SwiftTrans built on models like Qwen2.5-7B outperforms state-of-the-art baselines, including GPT-5.

**Strengths:**

1. This paper is well-motivated since the efficiency of code translation is practical and matters a lot in our real life.
2. This paper’s ideas are easy to follow, holistic, and pretty effective.
3. This paper contribute a new benchmark *SWIFTBENCH*, which takes account for the efficieny for code translation.

**Weaknesses:**

### **Multi-Perspective Exploration**
The idea of leveraging parallel ICL to generate diverse candidates is effective but not novel. I’m curious how well does the model perform with the direct use of ICL only (prompt) compared to your Hierarchical Guidance (training). I think this can be an important ablation study.

### **Difference-aware Selection**
I believe the motivation and ablation study for the Difference-aware Selection is insufficient. Firstly, given the extensive context windows of modern LLMs (e.g., Qwen2.5-3B supports 32k tokens), it’s reasonable to consider evaluating all `m` candidates **simultaneously** within a single, long context (or a batch of candidates). In this case the efficiency will be dramastically improve. Secondly, the selector relies on the `diff(tgt1, tgt2)` operation. This asymmetric presentation may introduce an inherent bias since `tgt1` presents full code but `tgt2`  only shows a partial (the diff with respect to `tgt1`). Therefore, the model might develop a preference for `tgt1` or `tgt2`. It’s recommanded to quantify the consistency of the selector.

Overall, I think the main weakness is the lack of comprehensive ablation study to prove the efficiency of individual parts of the framework (since the overall framework is complex).

**Questions:**

1. Can SWIFTTRANS be a purely inference-time framework (no training needed)? If yes, how does it perform?
2. How do you ensure the quality of SWIFTBench? Do you have a more comprehensive comparison of SWIFTBench with CodeNet and F2SBench since I think their sources are similar.
3. What's the efficiency of the whole framework? Do not have a longer inference time compared to F2STrans?

---

> ### Author Response · Authors · 2025-11-22
> **Response to Reviewer sJd7 (Part 1/3)**
>
> **We greatly appreciate the reviewer's insightful and constructive feedback, and we have carefully addressed each point in our response to resolve your concerns.
> Should any further issues remain, please feel free to share your additional comments, and we will continue actively responding to your comments and improving our submission.**
>
> ---
>
> >**Q1.**
> About Multi-Perspective Exploration: I’m curious how well does the model perform with the direct use of ICL only (prompt) compared to your Hierarchical Guidance (training).
> I think this can be an important ablation study.
>
> **A1.**
> This ablation is already included in our paper (**Figure 4b**).
> When no training is applied (i.e., directly prompting the raw Qwen2.5-3B model), the performance is as follows:
>
> |                           | Computational Accuracy (CA) ↑ | Execution Time (ET) ↓ |
> |:-------------------------:|:-----------------------------:|:---------------------:|
> | Qwen2.5-3B (ICL only; no training) | 36.5%                         | 1038 ms               |
> | Qwen2.5-3B + Hierarchical Guidance (trained)  | **88.9%** (+52.4%)     | **448 ms** (-590 ms)  |
>
> ---
>
> >**Q2.**
> About Difference-aware Selection:
> Given the extensive context windows of modern LLMs (e.g., Qwen2.5-3B supports 32k tokens), it’s reasonable to consider evaluating all $m$ candidates simultaneously within a single, long context (or a batch of candidates).
> In this case the efficiency will be dramastically improve.
>
>
> **A2.**
> We agree that evaluating all $m$ candidates within a single long context (i.e., a list-wise selection approach) can significantly improve inference efficiency.
> However, our experiments show that **list-wise selection consistently underperforms our pair-wise strategy** on code translation tasks.
>
>
> In our evaluation, we first use the Qwen2.5-3B-based MpTranslator to generate 10 candidate translations for each source program in CodeNet, F2SBench, and SwiftBench.
> We then apply different selection variants to these candidates.
> As shown in the table below, the pair-wise strategy outperforms the list-wise strategy across all settings—whether or not the selector is trained, and even when using a larger LLM (Qwen3-Next-80B) as the selector.
>
>
> | Selection Strategy | Trained? | LLM            | Computational Accuracy ↑   | Execution Time ↓  |
> |:------------------:|:--------:|:--------------:|:-----:|:------:|
> | Pair-wise          | No       | Qwen2.5-3B     | 84.6% | 482 ms |
> | List-wise          | No       | Qwen2.5-3B     | 83.8% | 524 ms |
> | Pair-wise          | Yes      | Qwen2.5-3B     | **86.9%** | **339 ms** |
> | List-wise          | Yes      | Qwen2.5-3B     | 86.3% | 357 ms |
> | Pair-wise          | No       | Qwen3-Next-80B | 86.3% | 403 ms |
> | List-wise          | No       | Qwen3-Next-80B | 86.0% | 420 ms |
>
>
> ---
>
> >**Q3.**
> The selector relies on the diff(tgt1, tgt2) operation.
> This asymmetric presentation may introduce an inherent bias since tgt1 presents full code but tgt2 only shows a partial (the diff with respect to tgt1).
> Therefore, the model might develop a preference for tgt1 or tgt2.
> It’s recommanded to quantify the consistency of the selector.
>
> **A3.**
> We acknowledge that a naive `diff(tgt1, tgt2)` formulation may introduce asymmetry.
> However, both our **training objective** and **empirical evaluation** explicitly address this issue.
>
>
> #### **(1) Bi-Judge loss mitigates asymmetry at the training level.**
>
> As described in **Section 2.2.2**, the **Bi-Judge loss** forces the selector to evaluate the pair (`tgt1`, `tgt2`) in **both directions**, preventing it from relying on superficial positional bias and encouraging it to focus on the semantic differences between codes.
> This design directly reduces the preference toward either `tgt1` or `tgt2`.
>
>
>
>
> #### **(2) Selector consistency is quantitatively measured in Table 3.**
>
> We introduced the **Order Sensitivity (OS)** metric to evaluate consistency when the order of the two translations is reversed.
> A lower OS indicates greater robustness to the potential asymmetry raised in the question.
> The results show:
>
> | Selector Variant        | Computational Accuracy ↑   | Execution Time ↓       | Order Sensitivity ↓     |
> |:-----------------------:|:---------:|:----------:|:--------:|
> | Vanilla Selector (Qwen2.5-3B)         | 86.1%     | 609 ms     | 64.2%    |
> | DiffSelector w/o Bi-Judge    | 87.7%     | 497 ms     | 18.7%    |
> | **DiffSelector (full)** | **88.9%** | **448 ms** | **6.4%** |
>
>
> The OS drops from **64.2% → 18.7% → 6.4%**, demonstrating that Bi-Judge effectively resolves the consistency concern.

---

> ### Author Response · Authors · 2025-11-22
> **Response to Reviewer sJd7 (Part 2/3)**
>
> >**Q4.**
> Can SwiftTrans be a purely inference-time framework (no training needed)?
> If yes, how does it perform?
>
>
> **A4.**
> Yes. SwiftTrans can be used purely at inference time, without any training, and still yields substantial improvements over base LLMs.
> Applied directly to Qwen2.5-3B, Qwen2.5-7B, and Qwen3-Next-80B, inference-only SwiftTrans achieves the following gains on CodeNet:
>
> | | Computational Accuracy ↑ (%) | Execution Time ↓ (ms) |
> |:---:|:---:|:---:|
> | Qwen2.5-3B |  44.8  |    416     |
> |  Qwen2.5-3B + SwiftTrans |        59.9 ($\Delta$=+15.1)  |    365  ($\Delta$=-51)     |
> | Qwen2.5-7B |    65.4      |    402     |
> |  Qwen2.5-7B + SwiftTrans |       73.8 ($\Delta$=+8.4)  |   332 ($\Delta$=-70)      |
> | Qwen3-Next-80B |     78.0     |    371     |
> |  Qwen3-Next-80B + SwiftTrans |       86.9 ($\Delta$=+8.9)  |   291 ($\Delta$=-80)      |
>
>
> These results show that even without any training, SwiftTrans consistently improves both correctness and runtime efficiency across different backbone sizes.
>
> ---
>
> >**Q5.**
> How do you ensure the quality of SwiftBench?
>
> **A5.**
> Each SwiftBench example consists of three components: the source program, ten efficiency-critical test cases, and a baseline execution time for the target code.
>
> **Source code quality:**
> To avoid benchmark contamination, all source programs are drawn from recently released problems on platforms such as Codeforces (June–August 2025).
> We additionally curate source programs that are functionally correct but inefficient, so that SwiftBench captures translation scenarios where the source code is correct yet slow.
>
>
> **Test case and baseline execution time quality:**
> Three independent annotation teams generated candidate efficiency-critical test cases and conservative correctness-first translations for every source program.
> This produced a pool of 30 candidate test cases and three conservative translations per example.
> We then manually select ten diverse and challenging test cases as the final set.
> To establish the baseline runtime, all conservative translations are executed on these cases, and the slowest runtime is used as the baseline.
>
> This multi-stage curation and verification process ensures that SwiftBench is diverse, challenging, contamination-free, and reliable for evaluating both correctness and efficiency.
>
> ---
>
> >**Q6.**
> Do you have a more comprehensive comparison of SwiftBench with CodeNet and F2SBench since I think their sources are similar.
>
>
> **A6.**
> Yes, a detailed comparison is provided in **Appendix B** of the paper.
>
> SwiftBench differs from CodeNet and F2SBench in several important ways that make it a more comprehensive and realistic benchmark for evaluating code translation:
>
> 1. It uses recently submitted source programs (June–August 2025), reducing benchmark contamination.
> 2. It reflects real-world scenarios where the source code is functionally correct but inefficient.
> 3. It provides ten carefully curated efficiency-critical test cases per example.
> 4. It includes a baseline execution time for the target code, enabling meaningful runtime evaluation.
>
>
> The table below summarizes the release period of the source programs and the baseline execution time across benchmarks:
>
> |     Benchmark     |    Source Date   | Baseline Execution Time |
> | :---------------: | :--------------: | :---------------------: |
> |      CodeNet      |    Before 2021   |          423 ms         |
> |      F2SBench     |     Mid-2024     |          735 ms         |
> | SwiftBench (Ours) | June–August 2025 |         1362 ms         |
>
> Because SwiftBench intentionally includes source programs with performance issues, its baseline execution times are substantially higher.
> The distribution of inefficiency types is shown below:
>
> |       Inefficiency Category      | Percentage |
> | :------------------------------: | :--------: |
> |          Redundant Logic         |    28.2%   |
> | Underused Language/Library Tools |    25.3%   |
> |     Highly Complex Algorithms    |    19.5%   |
> |    Suboptimal Data Structures    |    18.6%   |
> |       Mathematical Overhead      |    8.4%    |
>
> Together, these characteristics make SwiftBench more challenging, and better aligned with real-world code translation settings than CodeNet and F2SBench.

---

> ### Author Response · Authors · 2025-11-22
> **Response to Reviewer sJd7 (Part 3/3)**
>
> >**Q7.**
> What's the efficiency of the whole framework?
> Do not have a longer inference time compared to F2STrans?
>
>
> **A7.**
> SwiftTrans does introduce additional inference time compared to F2STrans, but this overhead is both **limited** and **fully controllable**.
> Two points are important:
>
> 1. **Candidate generation is fully parallelizable**, so using more candidates does not linearly increase wall-clock time.
> 2. In **single-translation mode** (i.e., generating one translation without invoking the selector), SwiftTrans has **the same inference cost as F2STrans**, while already outperforming it in both correctness and runtime efficiency.
>
> The table below reports the average per-sample inference time and translation quality on CodeNet, F2SBench, and SwiftBench.
> Here, “n-candidate mode” corresponds to generating **n candidate translations** before selection.
>
> | Method                         | Framework Inference Time ↓ | Computational Accuracy ↑ | Execution Time ↓ |
> |:------------------------------:|:------------------------:|:------------------------:|:----------------:|
> | F2STrans                       | 3.67 s                   | 82.0%                    | 744 ms           |
> | SwiftTrans (1-candidate mode)  | 3.88 s                   | 83.4%                    | 542 ms           |
> | SwiftTrans (10-candidate mode) | 6.04 s                   | 86.9%                    | 339 ms           |
>
> Overall, SwiftTrans offers substantial gains in correctness and runtime efficiency, and the inference overhead can be minimized—or eliminated—by using the single-candidate mode when required.

---

> > ### Comment · Reviewer_sJd7 · 2025-11-23
> >
> > Thanks for your comprehensive response. I think most of my questions and concerns are addressed. Therefore, I raise my score to 6.

---

> > > ### Author Response · Authors · 2025-11-23
> > > **Thank You for Your Follow-up Feedback**
> > >
> > > **Thank you very much for your thoughtful feedback and for raising your score to 6!**
> > > We deeply appreciate the time and effort you have invested in carefully reviewing our work and providing such constructive suggestions.
> > > This level of engagement is not only extremely helpful for us as authors, but also profoundly encouraging.
> > >
> > >
> > > In a year with a very high submission volume, it is clear that maintaining such a high standard of reviewing takes real commitment.
> > > Your detailed feedback reminded us of the positive impact that dedicated reviewers can have on the community.
> > > It is exactly this kind of careful and responsible reviewing that keeps conferences like ICLR strong and helps authors continue improving their work.
> > >
> > > We will incorporate the additional clarifications and results you suggested into the camera-ready version.
> > > Once again, thank you for recognizing our efforts.
> > > We hope that our work can contribute meaningfully to the code translation field.

---

### Official Review · Reviewer_gqC2 · 2025-10-31

**Soundness:** 3
**Presentation:** 3
**Contribution:** 3
**Rating:** 8
**Confidence:** 4

**Summary:**

The paper starts from the observation that hand-crafted translations are faster on average than the translations produced by LLM translators--even if those translations are correct (in terms of i/o-equivalence). It then presents a novel code-to-code translation approach with the goal of producing translations that are not only correct but also efficient.

To this end the authors employ a generator and a selector. The generator produces translations conditioned on the source language code and sets of exemplars of efficient code translations. The selector is prompted with a diff of two possible translations and picks the better one, considering both correctness and efficiency. The selector can be employed on an arbitrarily large set of candidate translations by performing pairwise comparisons in a bubblesort like manner. Both generator and selector are finetuned for this task.
To test this setup the authors extend CodeNet and F2SBench with new tests that increase the input size. In addition, a new dataset is created from Codeforces examples that intentionally introduces inefficient elements into the source language. The generator-selector approach yields significant improvements on all three datasets, to the point where small-scale LLMs with this approach outperform the much larger GPT5.

**Strengths:**

Exploring secondary characteristics of translation quality, such as efficiency, is an important research direction, and becomes more important as the state-of-the-art for correctly translating code improves. The authors do a good job quantifying this motivation with their preliminary study.

Furthermore, the presented approach yields impressive results (especially in comparison to GPT5) and significantly furthers the state-of-the-art.

Methodologically the paper is primarily a well-executed application of existing ideas; In-context learning, task finetuning, and LLM-as-a-judge--are all well established. However, there is value in how exactly these components are implemented. All parts of the setup appear well-motivated (e.g. to avoid diversity collapse) and the authors thoroughly ablate each component.
The augmented versions of CodeNet and F2SBench, as well as SwiftBench, provide a meaningful point of comparison for evaluation and are certainly of value for future work in this area.

Overall, the presentation quality is high; the paper presents a clear thread from motivation to method to evaluation. All figures are well done and aid understanding.

**Weaknesses:**

The paper leaves out some important details:

1. F2STrans was presented as a method to improve code style in code-to-code translations. What are the details of how it was applied to efficient translation?
2. How were the inefficiencies in SwiftBench created?
3. What is an "efficiency-critical" testcase? Is this just a larger input size?

The paper's language is in parts a bit too vague and too overdramatic. A more nuanced and grounded register, as well as a focus on the facts, would be more appropriate.

Examples of this include:
-	"[...] runtime efficiency has become as critical as functional correctness in evaluating program quality." (p. 1, ll. 14-16)
-	"efficiency-critical"
-	"code collected on online platforms (e.g., Codeforces)" (p. 4 ll. 169-170) As far as I understand only data from Codeforces is used.
-	"Multi-Perspective" I would argue different sets of demonstrations don't really constitute different perspectives.
Furthermore, I see two minor conceptual problems:
1. SwiftBench tests for a slightly different problem then the rest of the paper. The paper is motivated by the fact that an efficient implementation in the source language could be translated into something that is correct, but not efficient in the target language. SwiftBench evaluates a scenario, where the code in the source language is already inefficient. This is a different, and I believe less relevant, setting.
2. While it appears empirically successful, I fail to see why the number of demonstrations in the MpTranslator should be trained to correlate with target code efficiency.

The related work section misses the following important research directions in code to code translation:
-	Translations on intermediary representations of the code (e.g. [1], [2])
-	Rule based approaches (e.g. [3]) and neuro-symbolic hybrids (e.g. [4])

A final minor concern is the mention of the update frequency of SwiftBench (p.5 ll. 61-63) This is an unverifiable statement about the future and is of no value to the reader.

[1] Szafraniec et al., Code Translation with Compiler Representations

[2] Macedo et al., InterTrans: Leveraging Transitive Intermediate Translations to Enhance LLM-based Code Translation

[3] Galois, C2Rust URL https://www.galois.com/articles/c2rust

[4] Nitin et al., C2SaferRust: Transforming C Projects into Safer Rust with NeuroSymbolic Techniques

**Questions:**

- How was F2STrans applied to this setting?
- What are the details of the "efficiency-critical" test cases?
- How were the inefficiencies in SwiftBench created?
- Do you think this approach remains relevant as the baseline translation models become better?

---

> ### Author Response · Authors · 2025-11-22
> **Response to Reviewer gqC2 (Part 1/2)**
>
> **We thank the reviewer for the insightful and valuable comments.**
>
> ---
>
> >**Q1.** F2STrans was presented as a method to improve code style in code-to-code translations.
> What are the details of how it was applied to efficient translation?
>
> **A1.**
> F2STrans is designed primarily to improve the style of translated code, and we find that emphasizing stylistic refinement often comes at the cost of execution efficiency.
> For this reason, in our work F2STrans serves as an important baseline rather than an efficiency-oriented method.
>
> To ensure a fair comparison, we apply F2STrans exactly as described in its original paper when translating the programs in CodeNet, F2SBench, and SwiftBench.
> This includes using the same prompt format, temperature, and all other decoding configurations.
> We then evaluate the translated outputs and compare them with SwiftTrans in terms of functional correctness and runtime efficiency.
>
> ---
>
> >**Q2.**
> How were the inefficiencies in SwiftBench created?
>
> **A2.**
> We construct SwiftBench using correct but inefficient code submitted by developers on online programming platforms such as Codeforces.
> These programs are functionally correct but contain inefficiencies arising from redundant logic, suboptimal data structures, and other performance-hindering coding practices.
>
> The distribution of inefficiency types in SwiftBench is shown below.
> If a code snippet contains multiple inefficiencies, we categorize it by the dominant factor affecting performance.
>
> | Category | Percentage |
> |:----:|:----:|
> | Redundant Logic | 28.2% |
> | Underused Language/Library Tool | 25.3% |
> | Highly Complex Algorithm | 19.5% |
> | Suboptimal Data Structure | 18.6% |
> | Math Overhead | 8.4% |
>
>
>
> ---
>
> >**Q3.**
> What is an "effciciency-critical" testcase?
> Is this just a larger input size?
>
>
> **A3.**
> “Efficiency-critical” does **not** simply mean “larger inputs”.
> An input can be efficiency-critical even **without increasing size**—its **structure alone** may induce worst-case behavior.
> For example, a nearly-sorted or adversarial array can trigger worst-case performance in certain sorting implementations, and a small but extremely dense graph can stress graph algorithms with inefficient adjacency handling.
> Thus, efficiency-criticality arises from **structural difficulty**, not scale.
>
> ---
>
> >**Q4.**
> SwiftBench tests for a slightly different problem then the rest of the paper.
> The paper is motivated by the fact that an efficient implementation in the source language could be translated into something that is correct, but not efficient in the target language.
> SwiftBench evaluates a scenario, where the code in the source language is already inefficient.
> This is a different, and I believe less relevant, setting.
>
>
> **A4.**
> As noted in Lines 52–53 of the paper, conventional translation systems often not only introduce new inefficiencies but also carry over inefficiencies in the source code.
> While existing benchmarks such as CodeNet and F2SBench typically assume optimal source implementations, real-world code is often correct but suboptimal, and such inefficiencies readily propagate during translation.
>
>
> SwiftBench is designed to capture this overlooked yet practical scenario by evaluating cases where the source code itself is inefficient.
> It complements our main experiments by testing whether models can handle such inputs robustly, rather than relying on idealized source programs.
>
> ---
>
> >**Q5.**
> While it appears empirically successful, I fail to see why the number of demonstrations in the MpTranslator should be trained to correlate with target code efficiency.
>
>
> **A5.**
> As stated in Lines 195–197 of the paper, our intention is to use the number of demonstrations as **an explicit control signal for the desired level of optimization**.
>
> Functional correctness is the primary requirement in code translation.
> For complex source programs, using fewer demonstrations signals the model to emphasize correctness before attempting deeper efficiency improvements.
> Conversely, supplying more demonstrations provides a stronger optimization cue, prompting the model to perform more aggressive efficiency-oriented refinements.
> This design allows the model to modulate its optimization behavior in a controllable manner rather than relying on implicit heuristics.

---

> ### Author Response · Authors · 2025-11-22
> **Response to Reviewer gqC2 (Part 2/2)**
>
> >**Q6.**
> Do you think this approach remains relevant as the baseline translation models become better?
>
> **A6.**
> Yes, our SwiftTrans will remain relevant as base translation models continue to improve, precisely because it is model-agnostic and provides benefits that are complementary to stronger backbones.
>
> As shown in the table below, SwiftTrans built on a smaller backbone (Qwen2.5-3B) already surpasses the Qwen2.5-7B-based F2STrans, and scaling to Qwen2.5-7B further enables SwiftTrans to outperform GPT-5 in correctness while simultaneously improving execution efficiency.
> In contrast, simply scaling F2STrans from 3B to 7B yields virtually no efficiency improvement.
> These results suggest that stronger backbone models alone do not fully resolve the inherent tension between correctness and runtime efficiency, whereas our multi-perspective exploration combined with difference-aware selection offers orthogonal and consistently scalable gains on top of better models.
>
> | Method | LLM |Computational Accuracy (CA) ↑ | Execution Time (ET) ↓ |
> |:---:|:---:|:---:|:---:|
> | Correctness-Only | Qwen3-Next-80B |           73.3 %   |   776 ms      |
> | Correctness+Efficiency | Qwen3-Next-80B |    68.6 %   |  667 ms      |
> | Correctness→Efficiency | Qwen3-Next-80B |     59.3 %   |  565 ms      |
> | Correctness-Only | GPT-5 |                 86.4 %   |  528 ms      |
> | Correctness+Efficiency | GPT-5 |             79.8 %   |  453 ms      |
> | Correctness→Efficiency | GPT-5 |             62.7  %   |  399 ms      |
> |  F2STrans | Qwen2.5-3B |                     82.0 %   |  744 ms      |
> |  F2STrans | Qwen2.5-7B |                     84.6 %   |  731 ms      |
> | **SwiftTrans (Ours)** | Qwen2.5-3B |     86.9 %   |  339 ms      |
> | **SwiftTrans (Ours)** | Qwen2.5-7B |     **90.2 %**   |    **292 ms**    |

---

### Official Review · Reviewer_hiC3 · 2025-11-01

**Soundness:** 3
**Presentation:** 2
**Contribution:** 3
**Rating:** 6
**Confidence:** 4

**Summary:**

This work tackled the challenge of generating not only functional correct but also efficient code. They propose two modules: MpTranslator, a multi-perspective exploration technique for generating diverse translation candidates, and DiffSelector, a comparison framework to find the best candidate. These modules contribute to SWiftTrans, a code translation framework focusing on correctness and efficiency. For Multi-Perspective exploration, given a source code, an in-context learning phase that generated multiple sets of demonstrations for a given library for parallel translation. To help LLMs learn better in code translation, given multiple demonstrations, authors provide hierarchical data construction to filter the set of correct programs measured by test cases, and these programs have to be consistently 10% speedup from multiple rounds of code generation. The DiffSelect module leverages a difference-aware approach for candidate selection to avoid too similar target codes. The authors optimized the comparison process by applying the Bubble sort algorithm. Next, another round of optimization was performed to rank candidates based on the proposed judge loss function. In this evaluation, this work constructs SwiftBench, a dataset derived from the CodeNet dataset that includes more information about efficiency-critical test cases and a baseline execution time for the target code. By Computational Accuracy and Execution Time, the authors demonstrate that their proposed models perform best with Qwen2.5-7 B and outperform the existing well-known LLM, GPT-5, in code translation.

**Strengths:**

- The proposed problem is important.
- The design selections for two modules MpTranslator and DiffSelector are sound.
- This work proposes a rigorous process of evaluation with experiments.

**Weaknesses:**

- In terms of the paper written, there is a concept that was very unclear to me: “demonstration”. Also, there is lack of examples of Library C (mentioned in L152). Currently, I understand the demonstrations as the set of generated code by open LLMs for hierarchical data construction.
- In section 2.2.2, the authors mentioned in Line 227 that DiffSelector needs to rank incorrect translation code. However, in the earlier process of this framework, hierarchical data construction, functional incorrect codes were eliminated (Line 175-176), which is contradictory.
- Experiments lack configurations of building SwiftTrans over other open LLMs besides Qwen. Although it doesn’t mean the paper is invalid, a brief explanation about the decision to choose Qwen for optimization is needed.

**Questions:**

- Authors should clarify the definition of demonstrations and the demonstration library in the camera-ready version.
- An explanation of the reasons for choosing Qwen.
- Can this work be extended to ensure the generated code follows other perspectives besides functional correctness (such as coding style, code readability defined in [1]?

1.CodeUltraFeedback: An LLM-as-a-Judge Dataset for Aligning Large Language Models to Coding Preferences. Martin Weyssow, Aton Kamanda, Xin Zhou, Houari Sahraoui

---

> ### Author Response · Authors · 2025-11-22
> **Response to Reviewer hiC3 (Part 1/2)**
>
> We thank the reviewer for the insightful and valuable comments.
> We respond to each comment as follows and sincerely hope that our rebuttal could properly address your concerns.
>
>
> ---
> >**Q1.**
> In terms of the paper written, there is a concept that was very unclear to me: “demonstration”.
> Also, there is lack of examples of Library $\mathcal{C}$ (mentioned in L152).
>
> **A1.**
> Consistent with prior work on in-context learning [1,2,3], we use the term "demonstrations" to refer to the input–label pairs placed in the prompt.
> In our setting, each demonstration is a pair consisting of a source code snippet and its corresponding optimized target code.
> The “demonstration library $\mathcal{C}$” is simply a large collection of such demonstrations.
> As stated in Lines 153–154 and 179–180 of the paper, this library $\mathcal{C}$ is derived from our hierarchical data construction process.
>
>
> For example, a demonstration from the library $\mathcal{C}$ for the C-to-Python task is shown below:
>
> **Source code (C):**
> ```c
> int calculate_sum(int n) {
>     int sum = 0;
>     for (int i = 1; i <= n; i++) {
>         sum += i;
>     }
>     return sum;
> }
>
> int main() {
>     int n;
>     scanf("%d", &n);
>     printf("%d\n", calculate_sum(n));
>     return 0;
> }
> ```
>
>
> **Optimized target code (Python):**
> ```python
> def calculate_sum(n):
>     return n * (n + 1) // 2 # mathematically optimized for efficiency
>
> n = int(input())
> print(calculate_sum(n))
> ```
>
> [1] Qingxiu Dong, Lei Li, Damai Dai, Ce Zheng, Jingyuan Ma, Rui Li, Heming Xia, Jingjing Xu, Zhiyong Wu, Baobao Chang, Xu Sun, Lei Li, and Zhifang Sui. 2024. A Survey on In-context Learning. In Proceedings of the 2024 Conference on Empirical Methods in Natural Language Processing, pages 1107–1128, Miami, Florida, USA. Association for Computational Linguistics. (Google Scholar citations:2694)
>
> [2] Sewon Min, Xinxi Lyu, Ari Holtzman, Mikel Artetxe, Mike Lewis, Hannaneh Hajishirzi, and Luke Zettlemoyer. 2022. Rethinking the Role of Demonstrations: What Makes In-Context Learning Work?. In Proceedings of the 2022 Conference on Empirical Methods in Natural Language Processing, pages 11048–11064, Abu Dhabi, United Arab Emirates. Association for Computational Linguistics. (Google Scholar citations: 1838)
>
> [3] Xiaonan Li, Kai Lv, Hang Yan, Tianyang Lin, Wei Zhu, Yuan Ni, Guotong Xie, Xiaoling Wang, and Xipeng Qiu. 2023. Unified Demonstration Retriever for In-Context Learning. In Proceedings of the 61st Annual Meeting of the Association for Computational Linguistics (Volume 1: Long Papers), pages 4644–4668, Toronto, Canada. Association for Computational Linguistics. (Google Scholar citations: 197)
>
> ---
>
> >**Q2.**
> In section 2.2.2, the authors mentioned in Line 227 that DiffSelector needs to rank incorrect translation code.
> However, in the earlier process of this framework, hierarchical data construction, functional incorrect codes were eliminated (Line 175-176), which is contradictory.
>
>
> **A2.**
> The “hierarchical data construction” procedure is used to build the **training data for MpTranslator**.
> In contrast, Section 2.2.2 (“DiffSelector Optimization via Ordinal Guidance”) describes the **training process for DiffSelector**.
> These two procedures are independent.
>
> For DiffSelector, we deliberately construct three quality levels of candidate target-code translations:
> (1) efficient and correct translations,
> (2) correct but slower translations, and
> (3) incorrect translations.
> DiffSelector is trained to differentiate and rank these categories appropriately.

---

> ### Author Response · Authors · 2025-11-22
> **Response to Reviewer hiC3 (Part 2/2)**
>
> >**Q3.**
> Experiments lack configurations of building SwiftTrans over other open LLMs besides Qwen.
> Although it doesn’t mean the paper is invalid, a brief explanation about the decision to choose Qwen for optimization is needed.
>
> **A3.**
> Since Qwen is one of the most widely used open-source LLM families at present, our study primarily focuses on optimizing Qwen.
> The table below reports the average performance of SwiftTrans when applied to **Qwen2.5-3B**, **Qwen2.5-7B**, **StarCoder-7B**, and **DeepSeek-Coder-6.7B** on CodeNet, F2SBench, and SwiftBench.
> As shown, our method also improves StarCoder-7B and DeepSeek-Coder-6.7B, enabling them to surpass the surveyed baseline models.
>
> | Method | LLM |Computational Accuracy (CA) ↑ | Execution Time (ET) ↓ |
> |:---:|:---:|:---:|:---:|
> | Correctness-Only | Qwen3-Next-80B |           73.3 %   |   776 ms      |
> | Correctness+Efficiency | Qwen3-Next-80B |    68.6 %   |  667 ms      |
> | Correctness→Efficiency | Qwen3-Next-80B |     59.3 %   |  565 ms      |
> | Correctness-Only | GPT-5 |                 86.4 %   |  528 ms      |
> | Correctness+Efficiency | GPT-5 |             79.8 %   |  453 ms      |
> | Correctness→Efficiency | GPT-5 |             62.7  %   |  399 ms      |
> |  F2STrans | Qwen2.5-3B |                     82.0 %   |  744 ms      |
> |  F2STrans | Qwen2.5-7B |                     84.6 %   |  731 ms      |
> | **SwiftTrans (Ours)** | Qwen2.5-3B |     86.9 %   |  339 ms      |
> | **SwiftTrans (Ours)** | Qwen2.5-7B |     **90.2 %**   |    **292 ms**    |
> | **SwiftTrans (Ours)** | StarCoder-7B |   89.4 %   |     311 ms      |
> | **SwiftTrans (Ours)** | Deepseek-Coder-6.7B | 89.6%   |    296 ms      |
>
> ---
>
> >**Q4.** Can this work be extended to ensure the generated code follows other perspectives besides functional correctness (such as coding style, code readability defined in [4])?
>
> **A4.**
> Yes. Our work can be easily extended to additional dimensions of code quality.
> Although our current work focuses on functional correctness and execution efficiency, extending it to other aspects—such as memory usage or coding style—only requires minor prompt-level modifications in the multi-perspective exploration and difference-aware selection mechanisms.
>
> The table below shows the results of extending Qwen2.5-3B–based SwiftTrans to more code-quality dimensions.
> Beyond the Computational Accuracy and Execution Time metrics reported in the paper, we additionally include **Memory Usage** and the **Maintainability Index** [5], where the Maintainability Index provides a comprehensive assessment of code structure, readability, and overall maintainability.
>
>
> | Method | LLM | Computational Accuracy ↑ | Execution Time ↓ | Memory Usage ↓ | Maintainability Index ↑ |
> |:---:|:---:|:---:|:---:|:---:|:---:|
> |  F2STrans | Qwen2.5-3B |    82.0 %   |  744 ms      | 31.5 Mb | 23.5 |
> | **SwiftTrans (Ours)** | Qwen2.5-3B |     **86.9 %**   |  **339 ms**   |  **24.1 Mb** | **42.8** |
>
>
> [4] CodeUltraFeedback: An LLM-as-a-Judge Dataset for Aligning Large Language Models to Coding Preferences. Martin Weyssow, Aton Kamanda, Xin Zhou, Houari Sahraoui
>
> [5] Coleman D, Ash D, Lowther B, et al. Using metrics to evaluate software system maintainability[J]. Computer, 1994, 27(8): 44-49.

---

### Comment · Area_Chair_wVkZ · 2025-11-24
**Author Responses Are Ready - Please Review & Provide Feedback**

Dear Reviewers,

Thank you once again for your essential contributions to the review process. The authors have submitted their responses to your initial reviews.

I kindly ask you to carefully review the authors' responses for the papers you are handling. Your timely assessment of how the authors have addressed your original concerns is a critical step in reaching a final decision.

Please provide your feedback and any necessary updates to your reviews as soon as possible to ensure we can meet our tight schedule for the discussion phase.

Your prompt attention to this matter is highly appreciated.

Best regards,

Area Chair

---

### Author Response · Authors · 2025-12-03
**Paper Summary**

We sincerely thank the AC and all reviewers for their thoughtful evaluation of our work.
This paper focuses on improving both the functional correctness and runtime efficiency of translated code produced by current code translation systems.
The main contributions are as follows:

### **1. The SwiftTrans Framework**

To our knowledge, we are the first to systematically identify and address the runtime-efficiency issues of code generated by code translation systems.
Our SwiftTrans framework achieves high-quality translations through multi-perspective candidate generation and a difference-aware selection mechanism.

### **2. Extensions to Existing Benchmarks and the Construction of SwiftBench**

We extend existing benchmarks (CodeNet and F2SBench) and introduce **SwiftBench**, enabling comprehensive evaluation in terms of both correctness and efficiency.

### **3. Extensive Experimental Evaluation**

Experiments across eight benchmarks demonstrate consistent improvements over strong baselines.
These benchmarks span three levels of code translation:

* **Method/File level:** CodeNet, F2SBench, SwiftBench, PIE, xCodeEval
* **Class level:** ClassEval-T
* **Repository level:** AlphaTrans, RepoTransBench

---

> ### Author Response · Authors · 2025-12-03
> **Summary of Paper Revision**
>
> We have revised the paper based on the common concerns raised by the reviewers.
> The main updates are as follows:
>
> ---
>
> > **Q1. More comprehensive evaluation of translated code quality (e.g., memory usage)**
>
> *(Raised by Reviewers hiC3, 2b8R, avw2)*
>
> **A1.**
> **Table 4** of our paper has added memory usage and cyclomatic complexity to our evaluation.
> The results below show that SwiftTrans consistently improves functional correctness, runtime efficiency, memory usage, and structural simplicity:
>
> | Model | Functional Correctness ↑ (%) | Runtime ↓ (ms) | Memory ↓ (MB) | Cyclomatic Complexity ↓ |
> |:---------------------:|:---------------------:|:--------------:|:---------------:|:----------------:|
> | Qwen3-Next-80B        | 73.3                         | 776            | 27.6          | 6.5                     |
> | GPT-5                 | 86.4                         | 528            | 26.1          | 5.9                     |
> | F2STrans              | 84.6                         | 731            | 29.1          | 7.0                     |
> | **SwiftTrans (Ours)** | **90.2**                     | **292**        | **23.3**      | **5.1**                 |
>
> ---
>
> > **Q2. Inference efficiency of the SwiftTrans framework**
>
> *(Raised by reviewers sJd7, 2b8R, avw2)*
>
> **A2.**
> Although SwiftTrans generates multiple candidates, which increases inference time, two points are important:
>
> 1. **With a single candidate**, SwiftTrans remains competitive and outperforms baselines.
> 2. Candidate generation is **fully parallelizable**, so the additional overhead is limited.
>
> **Table 5** of our paper reports functional correctness and inference latency under different candidate counts.
> With 1 candidate, SwiftTrans is only 0.2s slower than F2STrans while improving correctness by 2.5%.
> Increasing candidates from 1 to 10 roughly doubles inference time but yields a 3.5% correctness gain.
>
> | Method          | Functional Correctness ↑ (%) | Inference Time ↓ (s) |
> | --------------- |:----------------------------:|:--------------------:|
> | Qwen3-Next-80B  | 73.3                         | 21.8                 |
> | GPT-5           | 86.4                         | 121.3                |
> | F2STrans        | 84.6                         | **5.1**              |
> | **SwiftTrans**  |                              |                      |
> | ├─ 1 candidate   | 87.1                         | 5.3                  |
> | ├─ 5 candidates  | 89.3                         | 8.1                  |
> | └─ 10 candidates | **90.2**                     | 10.2                 |
>
> ---
>
> > **Q3. Evaluation on class-level and repository-level benchmarks**
>
> *(Raised by Reviewer avw2)*
>
> **A3.**
> In addition to the original five method/file-level benchmarks, we have added experiments on **ClassEval-T** (class level) and **AlphaTrans** and **RepoTransBench** (repository level) in **Table 6**.
> The results are shown below:
>
> |                       | ClassEval-T | AlphaTrans | RepoTransBench |  CodeNet | F2SBench | SwiftBench |    PIE   | xCodeEval |
> | :-------------------: | :---------: | :--------: | :------------: | :------: | :------: | :--------: | :------: | :-------: |
> |     Qwen3-Next-80B    |     18.6    |    23.1    |       3.0      |   78.0   |   64.2   |    77.6    |   63.6   |    73.8   |
> |        F2STrans       |     21.6    |    16.6    |       0.0      |   89.8   |   75.9   |    88.1    |   89.2   |    89.1   |
> | **SwiftTrans (Ours)** |   **28.4**  |  **27.5**  |     **7.3**    | **94.8** | **82.6** |  **93.3**  | **92.3** |  **91.8** |
>
> These results demonstrate that SwiftTrans is effective across all three translation levels.
>
> ---
>
> > **Q4. Evaluation on more LLMs**
>
> *(Raised by reviewer hiC3)*
>
> **A4.**
> We have included additional results with **DeepSeek-6.7B** and **StarCoder-7B** (**Table 1** of the revised version).
> SwiftTrans remains effective across all tested models:
>
> |                                | Functional Correctness (%) ↑ | Runtime (ms) ↓ |
> | ------------------------------ | :--------------------------: | :------------: |
> | Qwen3-Next-80B                 |             73.3             |       776      |
> | GPT-5                          |             86.4             |       528      |
> | F2STrans (Qwen2.5-3B)          |             82.0             |       744      |
> | F2STrans (Qwen2.5-7B)          |             84.6             |       731      |
> | **SwiftTrans (Qwen2.5-3B)**    |           86.9           |     339    |
> | **SwiftTrans (Qwen2.5-7B)**    |           **90.2**           |     **292**    |
> | **SwiftTrans (DeepSeek-6.7B)** |           89.6           |     296    |
> | **SwiftTrans (StarCoder-7B)**  |           89.4           |     311    |

---

> > ### Author Response · Authors · 2025-12-03
> > **Rebuttal Summary**
> >
> > ## **Reviewer hiC3 (rating: 6)**
> >
> > | Weaknesses and Questions   | Summary of Our Response   |
> > | ---------- | ------- |
> > | Misunderstanding of the SwiftTrans workflow.                             | Addressed.                                                    |
> > | Need to validate SwiftTrans on LLMs beyond Qwen models.                  | Added evaluations on DeepSeek-6.7B and StarCoder-7B; see **QA4 of "Summary of Paper Revision"**. |
> > | Insufficient evaluation of translated code quality (e.g., memory usage). | Addressed with more comprehensive metrics; see **QA1 of "Summary of Paper Revision"**.           |
> >
> > ---
> >
> > ## **Reviewer gqC2 (rating: 8)**
> >
> > | Weaknesses and Questions | Summary of Our Response |
> > | -------- | -------- |
> > | Questions about the construction and motivation of SwiftBench.    | Addressed.     |
> > | Questions about the training process of MpTranslator.             | Addressed.          |
> > | Whether SwiftTrans remains effective with stronger LLM backbones. | Addressed with experiments showing broad effectiveness. |
> >
> > ---
> >
> > ## **Reviewer sJd7 (initial rating: 4 → updated to 6 in 24 Nov 2025)**
> >
> > | Weaknesses and Questions | Summary of Our Response |
> > | ----- | --------- |
> > | Missing ablation on the Hierarchical Guidance process.                       | Shown in **Figure 4b** of the original paper.                                                                   |
> > | Comparison between list-wise selection and our pair-wise selection.          | We demonstrated that list-wise selection performs worse within SwiftTrans.                                  |
> > | Input-order sensitivity of the pair-wise selector.                           | Mitigated via our **Bi-Judge loss**; results in **Table 3** confirm the improvement.                                |
> > | How does SwiftTrans perform without training, when directly applied to LLMs? | The results show that SwiftTrans can enhance Qwen2.5-3B, Qwen2.5-7B, and Qwen3-Next-80B even without any training, demonstrating that the framework is effective out of the box. |
> > | How is the quality of SwiftBench ensured?                                    | **Section 3.1** explains that three independent annotation teams constructed the dataset.                       |
> > | Missing comparison between SwiftBench and CodeNet/F2SBench.                  | Provided in **Appendix B** of the original paper.                                                               |
> > | Missing comparison of inference cost between SwiftTrans and F2STrans.        | With one candidate, SwiftTrans has nearly identical inference time but higher translation quality; see **QA2 of "Summary of Paper Revision"**. |
> >
> > ---
> >
> > ## **Reviewer 2b8R (rating: 2)**
> >
> > | Weaknesses and Questions | Summary of Our Response |
> > | -------- | ----- |
> > | Insufficient evaluation of translated code quality (e.g., memory usage). | The code translated by SwiftTrans shows clear advantages in functional correctness, runtime efficiency, memory usage, and cyclomatic complexity; see **QA1 of "Summary of Paper Revision"**.                                               |
> > | Missing discussion of SwiftTrans inference cost.                         | With one candidate, SwiftTrans has nearly identical inference time but higher translation quality; see **QA2 of "Summary of Paper Revision"**.                                               |
> > | Missing clarification of the default candidate size.                     | Stated in **Lines 269 and 364** of the original paper (default = 10). |
> > | Missing performance results under different candidate sizes.             | Provided in **Figure 3** of the original paper.                       |
> >
> > ---
> >
> > ## **Reviewer avw2 (rating: 2)**
> >
> > | Weaknesses and Questions | Summary of Our Response |
> > | ---------- | ------- |
> > | CodeNet is unreliable due to limited test coverage.                      | We extended its test suite and also evaluate on eight benchmarks. |
> > | Insufficient evaluation of translated code quality (e.g., memory usage). | The code translated by SwiftTrans shows clear advantages in functional correctness, runtime efficiency, memory usage, and cyclomatic complexity; see **QA1 of "Summary of Paper Revision"**.                        |
> > | Concerns about intentional inefficiencies in SwiftBench.                 | All inefficiencies come from real developer submissions, not synthetic edits.                       |
> > | Missing discussion of SwiftTrans inference cost.                         | With one candidate, SwiftTrans has nearly identical inference time but higher translation quality; see **QA2 of "Summary of Paper Revision"**. |
> > | Missing evaluation on repository-level benchmarks.                       | Addressed with results on AlphaTrans and RepoTransBench; see **QA3 of "Summary of Paper Revision"**.                                   |

---

### Meta-Review · Area_Chair_Pi3r · 2025-12-23

**Summary:**

This paper aims to improve both the functional correctness and runtime efficiency in code translation systems.

Strengths:
(1) important problem (code efficiency). (2) Solid design of two modules (MpTranslator and DiffSelector). (3) rigorous evaluation process. (4) newly introduced benchmark taking into account the efficiency.

Weaknesses:
(1) Insufficient evaluation of translated code quality (such as model family, limited test coverage, etc.).
(2) writing clarity and statement rigor. (3) missing related works. (4) insufficient motivation and ablation study for the difference-aware selection. (5) limited novelty of leveraging parallel ICL. (6) omission of memory cost and overall computational overhead in efficiency evaluation.

**Reviewer Concerns:**

Some concerns, especially those related to empirical evaluations were addressed.

**Reviewer Scores:**

sJd7 increased the initial rating 4  to 6. The other reviewers remain split and it is not clear if they will change the scores.

---

### Decision · Program_Chairs · 2026-01-26

Reject